# Representation of male features in the female mouse accessory olfactory bulb, and their stability during the estrus cycle

**Oksana Cohen, Yoram Ben-Shaul***

Department of Medical Neurobiology, Institute for Medical Research Israel Canada, Faculty of Medicine, Jerusalem, Israel

## eLife Assessment

This **important** work substantially advances our understanding of how accessory olfactory bulb neurons respond to social odor cues across the estrous cycle, showing that responses vary with the strain and sex of the odor source but display no consistent differences between estrous and non-estrous states. It employs a unique electrophysiology preparation that activates the vomeronasal organ pump via electric stimulation, enabling precise recordings of accessory olfactory bulb cell responses to different chemosignals in anesthetized mice. Overall, the study presents **convincing** findings on the stability and variability of accessory olfactory bulb response patterns, indicating that while accessory olfactory bulb detects social signals, it does not appear to interpret them based on reproductive state. This work will be of interest to those studying olfaction, social behavior, reproductive cycles, and systems neuroscience more broadly.

**\*For correspondence:**
yoramb@ekmd.huji.ac.il

**Abstract** Most behaviors result from integration of external and internal inputs. For example, social behavior requires information about conspecifics and internal physiological states. Like many other mammals, female mice undergo a reproductive cycle during which their physiology and behavioral responses to males change dramatically: during estrus, they are more receptive to male mating attempts. A critical element in reproductive behavior is the investigative stage, which in mice and many other species, strongly relies on chemosensation. While the initial approach mostly involves the main olfactory system (MOS), once physical contact is established, the vomeronasal system (VNS) is engaged to provide information about potential partners' characteristics. Given the estrus-stage-dependent behavioral response, we asked whether representations of male features in the first brain relay of the VNS, namely, the accessory olfactory bulb (AOB), change during the cycle. To this end, we used a stimulus set comprising urine samples from males of different strains and virility levels, as well as from estrus and non-estrus females. The stimulus set was designed to reveal if response patterns of AOB neurons conform to ethologically relevant dimensions such as sex, strain, and particularly, male virility state. Using extracellular recordings in anesthetized female mice, we find that most ethological categories contained in our dataset are not overrepresented by AOB neurons, suggesting that early stages of VNS processing encode conspecific information efficiently. Then, comparing neuronal activity in estrus and non-estrus females, we found that overall, response characteristics at the single neuron and population levels remain stable during the reproductive cycle. The few changes that do occur are not consistent with a systematic modulation of responses to male features. Our findings imply that the AOB presents a stable account of conspecific features to more advanced processing stages.

## Introduction

To efficiently navigate the environment, social animals must derive information about conspecifics and evaluate it in light of their own state. One notable example is reproduction, which requires coordination of sexual behavior with internal physiology (*Lenschow and Lima, 2020*; *Yin and Lin, 2023*). In mice, females undergo a 4-to-5-day cycle, characterized by hormonal and physiological changes that affect the reproductive system and the females' responses to male stimuli (*Zinck and Lima, 2013*; *Gutierrez-Castellanos et al., 2022*). The cycle includes a fertile estrus/proestrus phase accompanied by receptive behavior toward males, and a non-estrus phase, during which males are avoided (*Nelson and Kriegsfeld, 2022*). Sexual behavior is often divided into appetitive, consummatory, and refractory stages (*Yin and Lin, 2023*; *Nelson and Kriegsfeld, 2022*), and the appetitive stage can be further divided into detection, approach, and investigation.

In many species, chemosensation is a central sensory modality that plays a key role in social interactions and is critical during the appetitive stage. In rodents and most vertebrates, information about conspecifics is obtained via two major chemosensory systems. The main olfactory system (MOS) plays many roles, and, among other functions, is required for social behaviors (*Liberles, 2014*; *Spehr et al., 2006*). The vomeronasal system (VNS), in contrast, appears dedicated to cues from other organisms and conspecifics in particular (*Mohrhardt et al., 2018*). An important functional distinction between the systems involves stimulus uptake. The MOS detects volatile cues and can thus sense distant objects. During social interactions, it is likely required to decide whether to approach or withdraw. In contrast, the VNS requires physical contact for efficient stimulus uptake (*Mohrhardt et al., 2018*; *Hamacher et al., 2024*; *Luo et al., 2003*). Once contact is made, the VNS can provide further information that is needed for more fateful behavioral decisions. Thus, the VNS is particularly important during the investigative part of the appetitive stage (*Keller et al., 2009*; *Keller et al., 2006*; *Oboti et al., 2014*).

The first brain stage of the VNS is the accessory olfactory bulb (AOB), and its principal neurons are known as mitral-tufted cells (AOB-MCs). AOB-MCs represent a key node, as they receive information from vomeronasal sensory neurons (VSNs) and project to downstream limbic stages (*Mohrhardt et al., 2018*). Previous studies have shown that AOB-MCs respond to various bodily secretions, including feces (*Doyle et al., 2016*), saliva (*Kahan and Ben-Shaul, 2016*), vaginal secretions (*Kahan and Ben-Shaul, 2016*), and most prominently, urine (*Ben-Shaul et al., 2010*; *Hendrickson et al., 2008*). These studies also revealed that AOB neurons show differential responses to urine from males vs. females (*Ben-Shaul et al., 2010*; *Hendrickson et al., 2008*; *Tolokh et al., 2013*), from mice of different strains (*Ben-Shaul et al., 2010*; *Tolokh et al., 2013*), and from females in different physiological states (*Kahan and Ben-Shaul, 2016*). Thus, the AOB provides critical information to guide social decisions, such as required during reproductive behavior.

At the behavioral level, it was shown that female mice prefer male over female stimuli (*Hellier et al., 2018*; *Moncho-Bogani et al., 2002*; *Martínez-Ricós et al., 2007*), dominant over nondominant males (*Coopersmith and Lenington, 1992*), and intact vs. castrated males (*Martínez-Ricós et al., 2008*). Here, to reveal how sensory representations of potential mating partners are represented in the AOB, we applied a stimulus set that includes urine from castrated, sexually naive, and proven-breeder dominant males (*Figure 1*). These stimuli correspond, respectively, to males that *cannot* breed, *may* be able to breed, and males that have bred successfully. Our stimulus set, therefore, represents varying levels of virility, allowing us to reveal if AOB-MCs represent male reproductive status.

Differential neuronal responses as a function of some property are often interpreted as 'tuning' to it. However, this is unwarranted unless selectivity is demonstrated across a diverse stimulus set, representing multiple sources of variability (*Rokni and Ben-Shaul, 2024*). For example, to demonstrate that a neuron is tuned to 'dominant males', selectivity should remain across a wide range of stimuli representing different instances of dominant male samples. Since the range of possible stimuli is unlimited, this goal can never be fully attained. Nevertheless, any expansion of the stimulus set allows a better characterization of response properties. Guided by these considerations, for each male virility state, we applied male urine stimuli from three distinct strains. This not only reduces the risk for misleading strain-specific conclusions, but also allows testing if AOB neurons are tuned to specific ethologically-relevant features embedded in our stimulus set. Revealing the response properties of AOB-MCs is particularly interesting given their dendritic fields, which can sample multiple glomeruli (*Takami and Graziadei, 1991*; *Larriva-Sahd, 2008*). This organization could give rise to

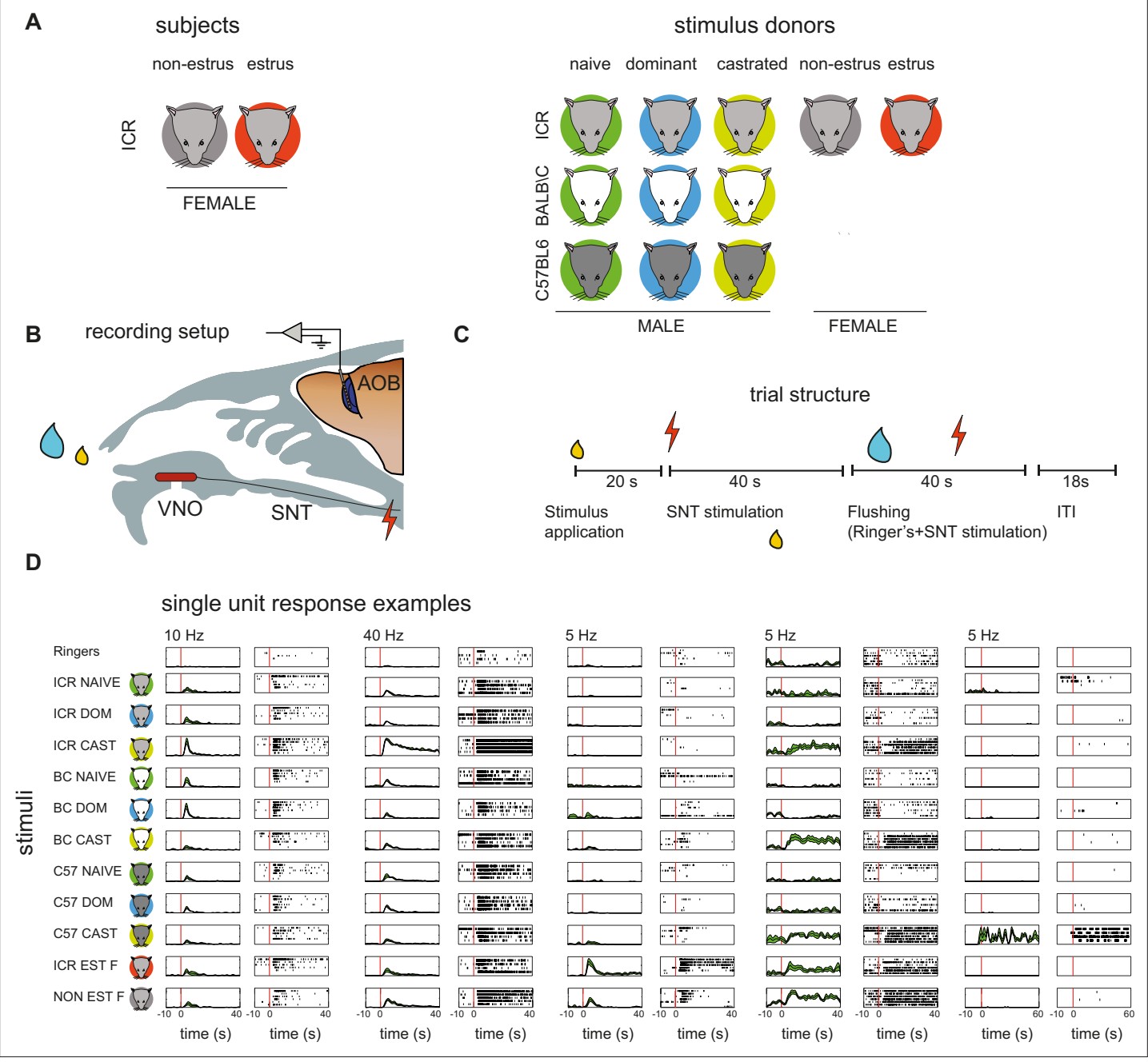

**Figure 1.** Experimental design and examples of single unit recordings. (**A**) Icons describing the different subject and stimulus donors. Mouse colors denote the strain, while background colors denote the status (estrus, non-estrus for females, naive, dominant, castrated for males). (**B**) Schematic of the recording setup. A multisite electrode probe is advanced to the external cell layer of the accessory olfactory bulb (AOB). Stimuli (yellow drop) are applied to the nostril and, after a delay, the sympathetic nerve trunk (SNT) is stimulated. Between stimulus presentations, the nasal cavity and VNO are flushed with Ringer's solution. VNO: vomeronasal organ. (**C**) Timeline of an individual trial. After a 20 s delay, the SNT is activated to induce VNO activation. Following another 40 s, the flushing procedure is initiated. An inter-trial interval (ITI) of 18 s is applied between consecutive stimulus presentations. During the experiment, stimuli are delivered repeatedly in blocks. Within each block, each of the individual stimuli is presented in a random order. (**D**) Examples of single unit responses to all stimuli. For each unit, responses to each of 11 stimuli are shown. Left panels show the mean firing rate (peri-stimulus time histogram [PSTH]), while right panels show corresponding raster displays. Green shadings in PSTHs indicate the standard error of the mean firing rate as a function of time. The red vertical line indicates SNT stimulation, except for the unit on the right, where it indicates stimulus application to the nostril. In this specific example, responses began immediately after stimulus application, prior to SNT stimulation.

response profiles corresponding to combinatorial molecular patterns and hence to ethologically relevant features.

As mentioned above, reproductive behavior requires evaluation of information about potential mating partners in light of the subject's own physiological states. Some studies (*Mossman and Drickamer, 1996*; *McCarthy et al., 2018*) (but see also *Moncho-Bogani et al., 2004*; *Nomoto and Lima, 2015*) indicate that not only the probability of mating, but also the initial preference toward male stimuli, is modulated by the estrus stage. Thus, in the context of female sexual behavior, it is expected that at some stage of processing, neuronal representations of male stimuli would depend on reproductive state. Consistent with this idea, it was shown that neuronal activity and/or connectivity in hypothalamic brain regions change during the female mouse estrus cycle (*Nomoto and Lima, 2015*; *Yin et al., 2022*; *Inoue et al., 2019*; *Wang et al., 2016*; *McHenry et al., 2017*; *Dias et al., 2021*; *Christensen et al., 2011*; *Knoedler et al., 2022*; *Liu et al., 2024*). Furthermore, response properties of VSNs are also altered during the estrus cycle (*Dey et al., 2015*; *Cherian et al., 2014*; *Eckstein et al., 2020*; *Stowers and Liberles, 2016*). However, it was not known if and how AOB sensory representations are modulated during the reproductive cycle. To this end, we recorded neuronal activity from naturally cycling female mice and examined how response properties of AOB-MCs change during the cycle. We hypothesized that responses to 'virile stimuli' will be somehow amplified during estrus, reflecting heightened sensitivity, or perhaps increased discriminatory ability.

Thus, this study addresses two interrelated goals. First, to reveal how male reproductive status is reflected by AOB neuronal activity. Second, to determine if representation of the males' reproductive state is altered during the course of the estrus cycle.

## Results

We recorded stimulus-induced responses from the AOB of estrus and non-estrus anesthetized adult ICR female mice (*Figure 1A and B*). Stimulus presentation includes sample application to the nostril, followed by electric stimulation of the sympathetic nerve trunk, which triggers stimulus uptake by the VNO (*Figure 1C*). During each session, we repeatedly present one of several stimuli in a pseudo-random order, with each stimulus presented at least four times (see Materials and methods for details). Between consecutive stimulus presentations, the VNO and nasal cavity were washed with Ringer's solution. The stimulus set includes urine from sexually naive, castrated, and sexually experienced dominant male mice, from each of three strains: BALB/C (BC), C57BL/6 (C57), or ICR (*Figure 1A*). These stimuli span two independent dimensions: male reproductive state and strain. We also presented urine from estrus and non-estrus ICR females (*Figure 1A*). Response magnitudes were quantified as the baseline-subtracted average firing rate, within a temporal window starting at stimulus application and ending 40 s after electrical stimulation of the sympathetic nerve trunk. Examples of single unit responses are shown in *Figure 1D*. The dataset includes 241 single units from 21 recording sites from 17 estrus females, and 305 single units from 28 recording sites from 20 non-estrus females.

### Population-level representations and general response features

Responses from all recording sessions (for both estrus and non-estrus females, n=546 single units) to each of the stimuli are shown in *Figure 2A* using a normalized response matrix, with nonsignificant responses (significance threshold of 0.05 using a nonparametric ANOVA, see Materials and methods) set to 0. As a population, AOB neurons respond to each of the stimuli in the set. To compare population activity patterns across stimuli, we applied the correlation distance metric and classical multidimensional scaling. This representation shows that the two female stimuli elicit similar patterns, distinct from all male stimuli (*Figure 2B*). Note that like all dimensionally reduced representations, this is an approximation. Here, it represents about 60% of the actual distances (the mean absolute error between the actual distances and the approximated distances is 0.29, whereas the mean of the original distances is 0.73). As for intact male stimuli, the main determinant of response similarity is strain (rather than physiological status): naive and dominant C57 stimuli form one cluster, while naive and dominant BC and ICR males form another cluster. Responses to castrated urine are distinct from male- or female-induced responses. Yet, here too, ICR and BC castrated urine are closer to each other than to C57 urine.

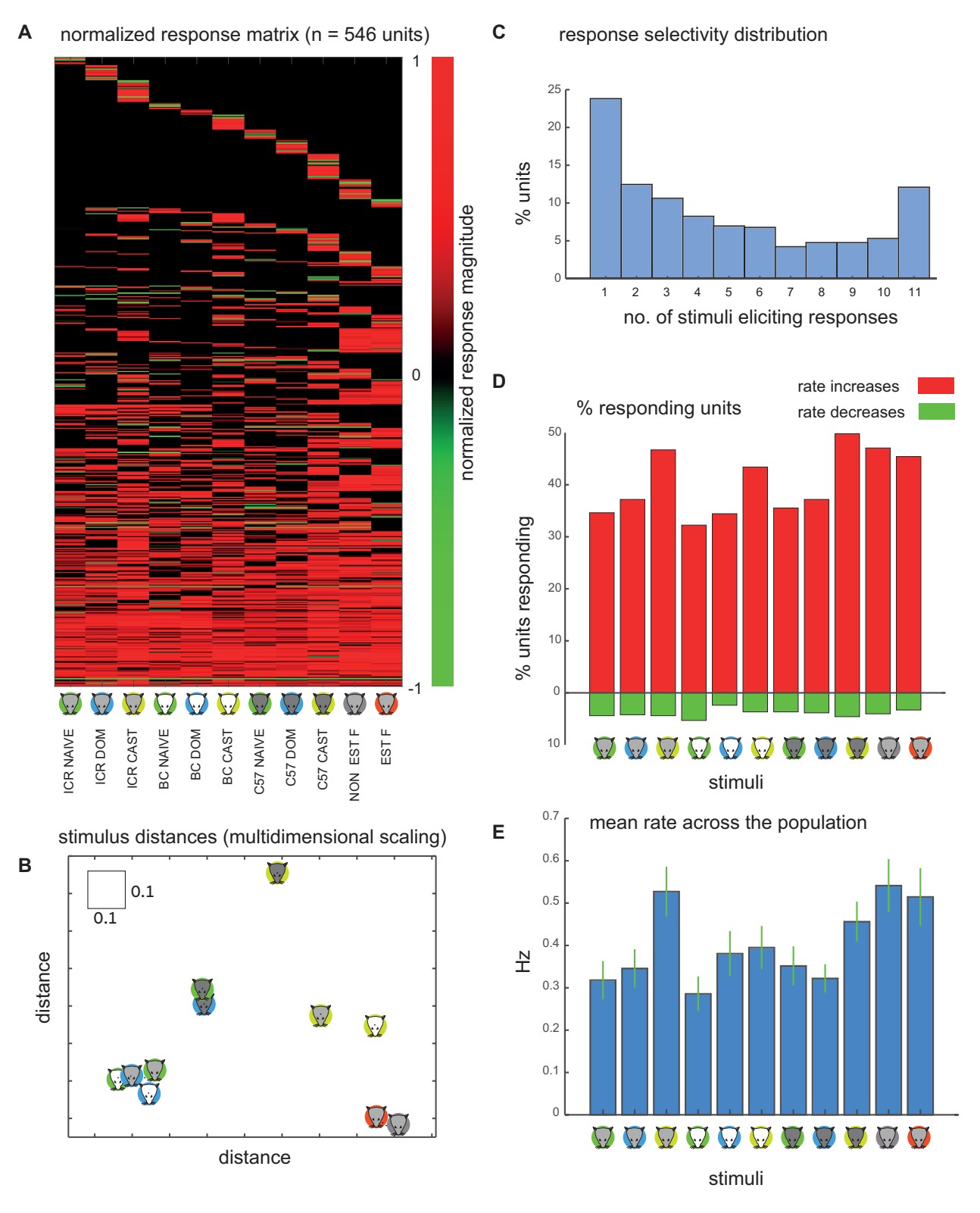

**Figure 2.** Basic response characteristics. (**A**) Colormap of responses. Each neuron (n=546) is represented by one row, and each stimulus by one column, as indicated at the bottom. Responses are normalized between –1 and 1. In this representation (and some analyses), nonsignificant responses are assigned a value of 0. (**B**) Classical multidimensional scaling of population-level responses using the data shown in panel A and a correlation distance metric. The mean error due to the reduced 2D representation is 0.29, whereas the mean of the original distances is 0.73. Note the square indicating

*Figure 2 continued on next page*

*Figure 2 continued*

a distance of 0.1 for comparison. (**C**) Distribution of the number of significant responses across all neurons. (**D**) Percentage of neurons responding to each of the stimuli, separately for rate increases (upper bars, red) and decreases (lower bars, green). (**E**) Mean response magnitude, across all recorded neurons to each of the stimuli. Green lines indicate the standard error of the mean. See *Table 1* for statistical comparison of stimulus magnitudes.

The distribution of the number of effective stimuli per neuron is shown in *Figure 2C*. Most neurons respond to only one stimulus, although a significant fraction responds to all stimuli. The percentage of neurons with significant rate increases and decreases, for each of the stimuli, is shown in *Figure 2D*. Consistent with previous studies in the AOB, rate increases are far more frequent than rate decreases (*Yoles-Frenkel et al., 2022*; *Bansal et al., 2021*). Also consistent with other studies, female stimuli activate more neurons than male stimuli (*Hendrickson et al., 2008*; *Bansal et al., 2021*; *Nodari et al., 2008*). Somewhat surprisingly, yet also consistent with previous work (*Yoles-Frenkel et al., 2022*), castrated stimuli are at least as effective as intact male stimuli.

Mean response strengths for each of the stimuli across the population are shown in *Figure 2E*. This value effectively combines the number of responsive neurons and their response magnitudes. This representation shows that female and castrated male stimuli elicit the strongest responses. A statistical comparison of response magnitudes is given in *Table 1*.

## Defining response patterns of AOB neurons

To better understand how behaviorally relevant features are represented by AOB neuronal activity, we next examine the response patterns of AOB neurons. Using stimuli from three strains and three male reproductive states, we define patterns corresponding to strain and to male virility status. Our stimulus set contains two female stimuli (estrus and non-estrus ICR urine), which allows us to define additional response patterns corresponding to sex and strain. We define response patterns using relative response magnitude (*Figure 3A*). For example, to fulfill a *dominant* pattern (DOM in *Figure 3A*), responses to all *dominant* male stimuli must be stronger than responses to any of the other stimuli. The complementary response pattern (designated as ~DOM) is equally informative. By setting a threshold, neurons with these response patterns suffice to determine if the stimulus source is a dominant male (*given* the present stimulus set). Note also that while we use this approach to define neuronal receptive fields, we are not suggesting that decoding is explicitly achieved in this manner. In addition, we defined 'strain-adjusted' and 'state-adjusted' response patterns (an example is shown in *Figure 3A*). The rationale behind these patterns is that some chemical feature may be modulated by both virility state and strain. For example, a *dominant strain-adjusted* pattern requires that for each strain, the response to the dominant stimulus will be stronger than the responses to the other stimuli *from the same strain*. Here too, we also considered complementary patterns, as they are also informative. We note that criteria for adjusted patterns are less stringent than for the standard patterns. Furthermore, some patterns are not mutually exclusive, and thus, a neuron may fulfill more than a single pattern.

## Most response patterns are not overrepresented

Responses of single neurons, sorted according to response types, are shown in *Figure 3B* (only basic and complementary response patterns are shown). The image shows that most neurons do not conform to any of the response patterns (78%, 433 of 557 total patterns; note that the number of patterns exceeds the number of neurons since a given neuron may conform to more than one pattern). While some patterns (e.g. *female*) are frequent, others are associated with only a few neurons (e.g. *naive* and *dominant* patterns). However, to gain insight about the sampling properties of AOB neurons, we asked which patterns are represented more than expected by chance. Here, chance implies that responses to each of the stimuli are independent of each other. Practically, we apply bootstrapping (with n=100,000 shuffles, see Materials and methods) to derive the frequencies for each of the patterns by chance. Bootstrapped distributions and the actual number of observed neurons for each of the patterns are shown in *Figure 3B*. Considering basic and reverse patterns, we find that *female*, *~female*, *C57 male*, and *castrated* are overrepresented (p<0.05, *Figure 3C*). For strain-adjusted or state-adjusted patterns (*Figure 3D*), we find overrepresentation of *C57/state*, *cast/strain*, and *~cast/strain*. Note that the observed frequencies are determined by the chemical profiles of the stimuli and the sampling properties of neurons. This idea and its implications are further developed in the

**Table 1.** Comparison of stimulus response magnitudes across stimulus pairs (related to *Figure 2E*).

**Significant median differences in mean responses between stimulus pairs, with Tukey-Kramer correction for multiple comparisons**

M_ICR_NAIVE vs M_ICR_CAST p: 0.042255

M_ICR_NAIVE vs M_C57_CAST p: 0.015033

M_ICR_NAIVE vs F_ICR_NON_EST p: 0.0010519

M_ICR_NAIVE vs F_ICR_EST p: 0.037127

M_ICR_CAST vs M_BC_NAIVE p: 0.0052463

M_BC_NAIVE vs M_C57_CAST p: 0.0014844

M_BC_NAIVE vs F_ICR_NON_EST p: 6.5788e-05

M_BC_NAIVE vs F_ICR_EST p: 0.004468

M_BC_DOM vs F_ICR_NON_EST p: 0.045776

**Significant median differences in mean responses between stimulus pairs without correction for multiple comparisons**

M_ICR_NAIVE vs M_ICR_CAST p: 0.0010636

M_ICR_NAIVE vs M_C57_CAST p: 0.00033956

M_ICR_NAIVE vs F_ICR_NON_EST p: 2.0771e-05

M_ICR_NAIVE vs F_ICR_EST p: 0.00091914

M_ICR_DOM vs M_ICR_CAST p: 0.030301

M_ICR_DOM vs M_C57_CAST p: 0.013284

M_ICR_DOM vs F_ICR_NON_EST p: 0.001636

M_ICR_DOM vs F_ICR_EST p: 0.027304

M_ICR_CAST vs M_BC_NAIVE p: 0.00011046

M_ICR_CAST vs M_BC_DOM p: 0.023567

M_ICR_CAST vs M_C57_DOM p: 0.033367

M_BC_NAIVE vs M_BC_CAST p: 0.025458

M_BC_NAIVE vs M_C57_NAIVE p: 0.046268

M_BC_NAIVE vs M_C57_CAST p: 2.9626e-05

M_BC_NAIVE vs F_ICR_NON_EST p: 1.2366e-06

M_BC_NAIVE vs F_ICR_EST p: 9.3289e-05

M_BC_DOM vs M_C57_CAST p: 0.010052

M_BC_DOM vs F_ICR_NON_EST p: 0.0011646

M_BC_DOM vs F_ICR_EST p: 0.021159

M_BC_CAST vs F_ICR_NON_EST p: 0.008915

M_C57_NAIVE vs M_C57_CAST p: 0.029007

M_C57_NAIVE vs F_ICR_NON_EST p: 0.0042801

M_C57_DOM vs M_C57_CAST p: 0.014786

M_C57_DOM vs F_ICR_NON_EST p: 0.001865

M_C57_DOM vs F_ICR_EST p: 0.030109

The tests were conducted using a nonparametric ANOVA (Kruskal-Wallis) and the multcompare MATLAB function with the Tukey-Kramer multiple comparisons correction (top) or without correction (bottom).

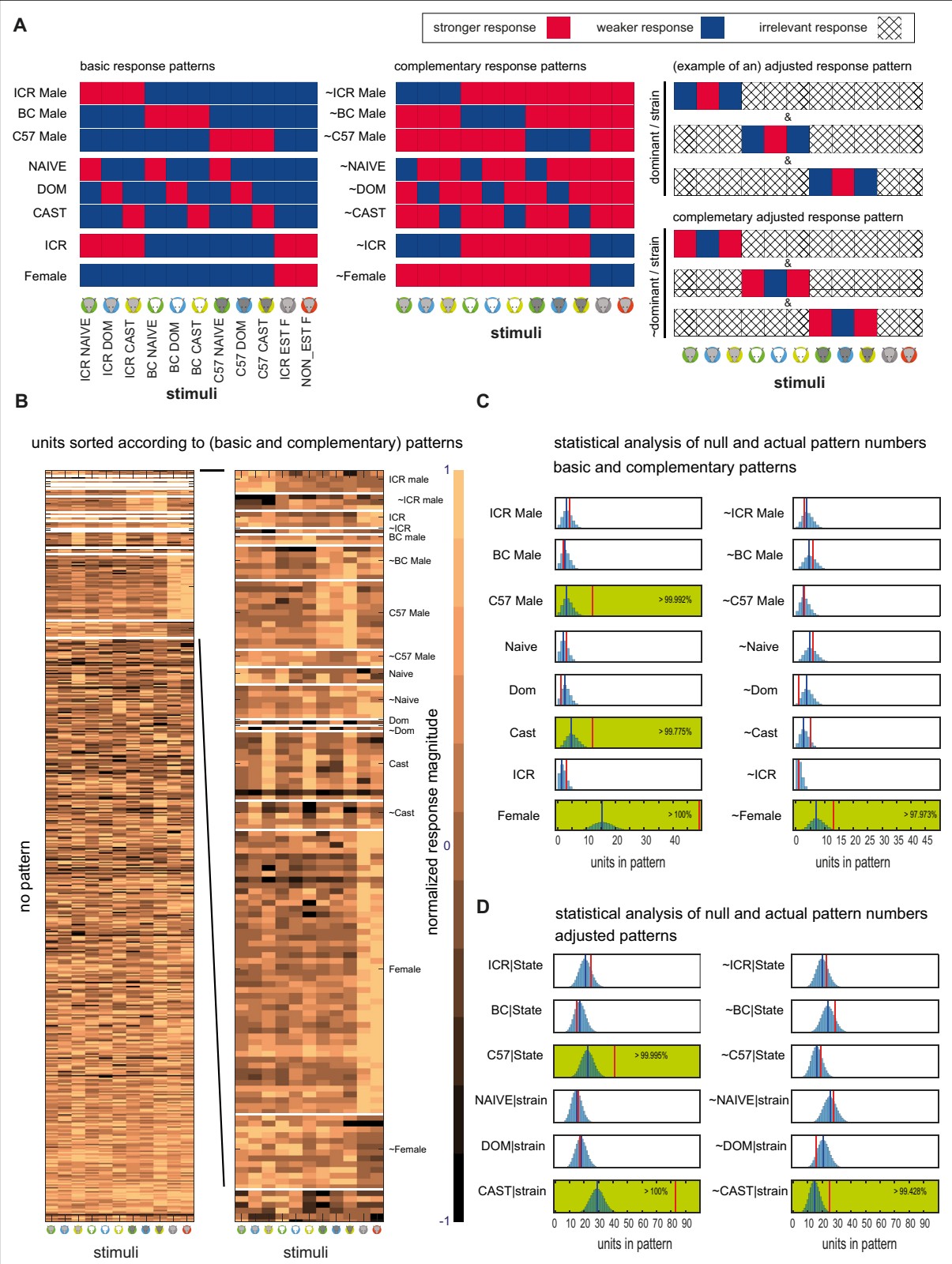

**Figure 3.** Analysis of response patterns. (**A**) Graphical representation of response pattern definitions. To fulfill a pattern, each of the stimuli indicated in red should elicit a stronger response than each of the stimuli shown in blue. The left and center panels show the basic response patterns and their complementary patterns, respectively. The panels on the right provide two examples of adjusted response patterns. For example, a dominant/strain pattern requires fulfillment of three conditions, so that within each strain, responses will be strongest to the dominant stimulus. Within a given condition,

*Figure 3 continued on next page*

*Figure 3 continued*

hatched squares indicate stimuli that are irrelevant for the pattern in question. The complementary-adjusted response patterns are not shown. (**B**) All response patterns across neurons. Note that a neuron may fulfill more than one pattern and thus may appear in more than one row. The right panel provides an expanded view of some response patterns. The adjusted categories are not shown in this representation. (**C**) Frequency analysis of each of the basic and complementary response patterns. The response pattern name is indicated next to each panel. Within each panel, the blue histogram shows the shuffled distribution of pattern frequencies (n=100,000 shuffles). The vertical blue lines show the mean value of the shuffled distribution. Red vertical lines indicate the actual number of observed patterns. Green shaded panels indicate significant cases (i.e. the estimated probability to obtain the observed number of cases by chance is less than 5%). In most cases, the probability is considerably lower (as indicated by the numbers within each panel). (**D**) Like C for adjusted patterns.

Discussion. Presently, we note that most of the male strain-specific patterns and the male-virility-state patterns are *not* more frequent than expected by chance.

## Neuronal representations are broadly stable over the estrus cycle

We next compare response properties of AOB neurons under estrus and non-estrus states, beginning with examination of responses at the population level. *Figure 4A* shows neuronal responses partitioned according to the estrus state of the subject (the same data as in *Figure 2A*). Each pair of stimuli (i.e. each pair of columns within the matrices in *Figure 4A*) is associated with a distance, which we quantify using two common metrics: correlation and Euclidean distances. Our 11-stimulus set yields 55 pairwise distances, which together provide a measure of representational space. The distance matrices for both states are shown in *Figure 4B* (correlation distance) and *Figure 4C* (Euclidean distance). Next, we compare the distances under both states, for both distance metrics (*Figure 4D and E*). We find a positive and highly significant correlation between representations under the two states. Thus, at the population level, representational space remains roughly stable across the estrus cycle. Nevertheless, several points in *Figure 4D and E* clearly diverge from the diagonal, implying that representations are not identical under the two states. We next examine these differences in more detail.

## Response intensity of most stimuli is stable across the estrus cycle

We next examine the intensity of specific stimuli across the cycle. Specifically, we ask if stimuli associated with virile males elicit stronger responses during estrus. For each stimulus and both states, we quantify response magnitudes using the fraction of responsive neurons and the mean response strength across the population. We compare the fraction of responses (*Figure 5A*) using a binomial exact test and the mean response strength (*Figure 5B*) using one-way nonparametric ANOVA. After (Bonferroni) correction for multiple comparisons, the only significant difference is that during estrus, the *fraction* of responsive neurons to castrated BC male urine increases. There are no significant differences in response *magnitude* to any of the stimuli. Consistent with this, a two-way ANOVA with stimulus and state as factors indicates that the only significant factor is stimulus (p-values were $1.5 \times 10^{-5}$, 0.7688, and 0.8193 for stimulus, state, and stimulus × state interaction, respectively). Thus, we conclude that there is no consistent change in response strength across the estrus cycle, and, in particular, no increase in response efficacy of stimuli associated with higher virility.

## The relative magnitude of response strengths remains stable across the estrus cycle

Although our previous analyses ruled out state-dependent changes in absolute response magnitudes, they did not directly test changes in *relative* response magnitude. For example, the relative strength of responses to dominant and naive stimuli might change as a function of the estrus state. To explore this scenario, and other related scenarios, we defined a preference index: $PI_{AB} = (R_A - R_B)/(R_A + R_B)$, where $R_A$ and $R_B$ are the responses to stimuli A and B, respectively. $PI_{AB}$ ranges between –1 (exclusive response to stimulus B) and 1 (exclusive response to stimulus A). After calculating these values for each neuron, we obtained their means across the population and compared them (55 preference indices, defined by our 11 stimuli dataset) under both states. The comparison (*Figure 5C*) reveals a positive (0.41) and significant (p=0.002) correlation between the two states.

Despite the similarity, some pairwise indices change and even reverse signs across the states (as indicated by points located in the upper-left and lower-right quadrants of *Figure 5C*). To reveal if these reflect meaningful changes in responsiveness to *male* features, we examined pairwise indices

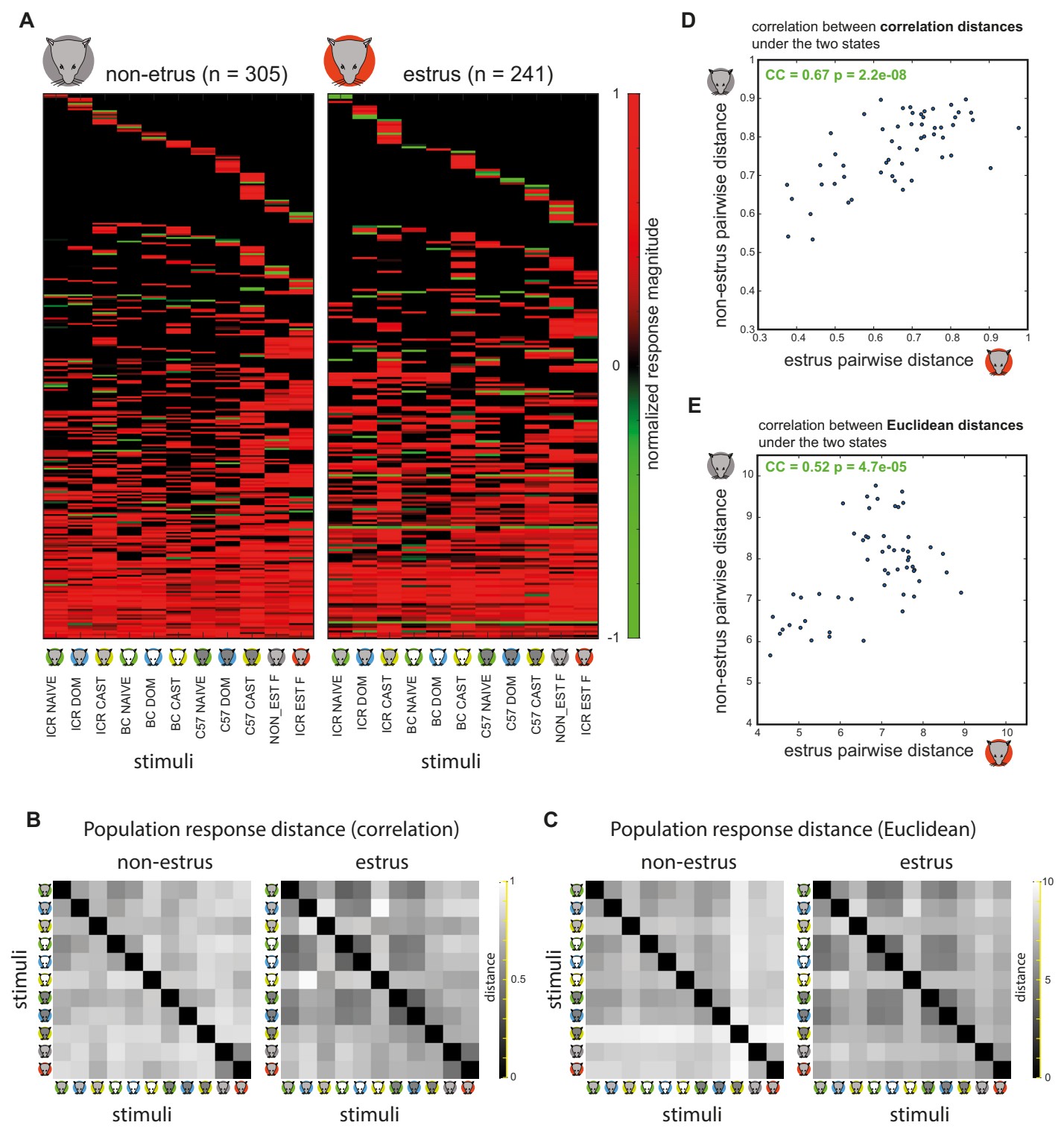

**Figure 4.** Comparison of population-level representations in estrus and non-estrus females. (**A**) Responses of individual neurons. Same data as in *Figure 2A*, divided by reproductive states. (**B**) Pairwise population response distances in non-estrus (left) and estrus (right) using the correlation distance measure. Each pixel represents the distance between two columns within a given matrix from panel A. By definition, these matrices are symmetric around the diagonal. (**C**) Same as in B, using the Euclidean distance measure. (**D**) Correlation of population-level pairwise distances across the two reproductive states, using the correlation distance measure. (**E**) Same as in D using the Euclidean distances. In both D and E, the correlation coefficient (CC) and the probability to obtain it by chance (p-value) are indicated within each panel.

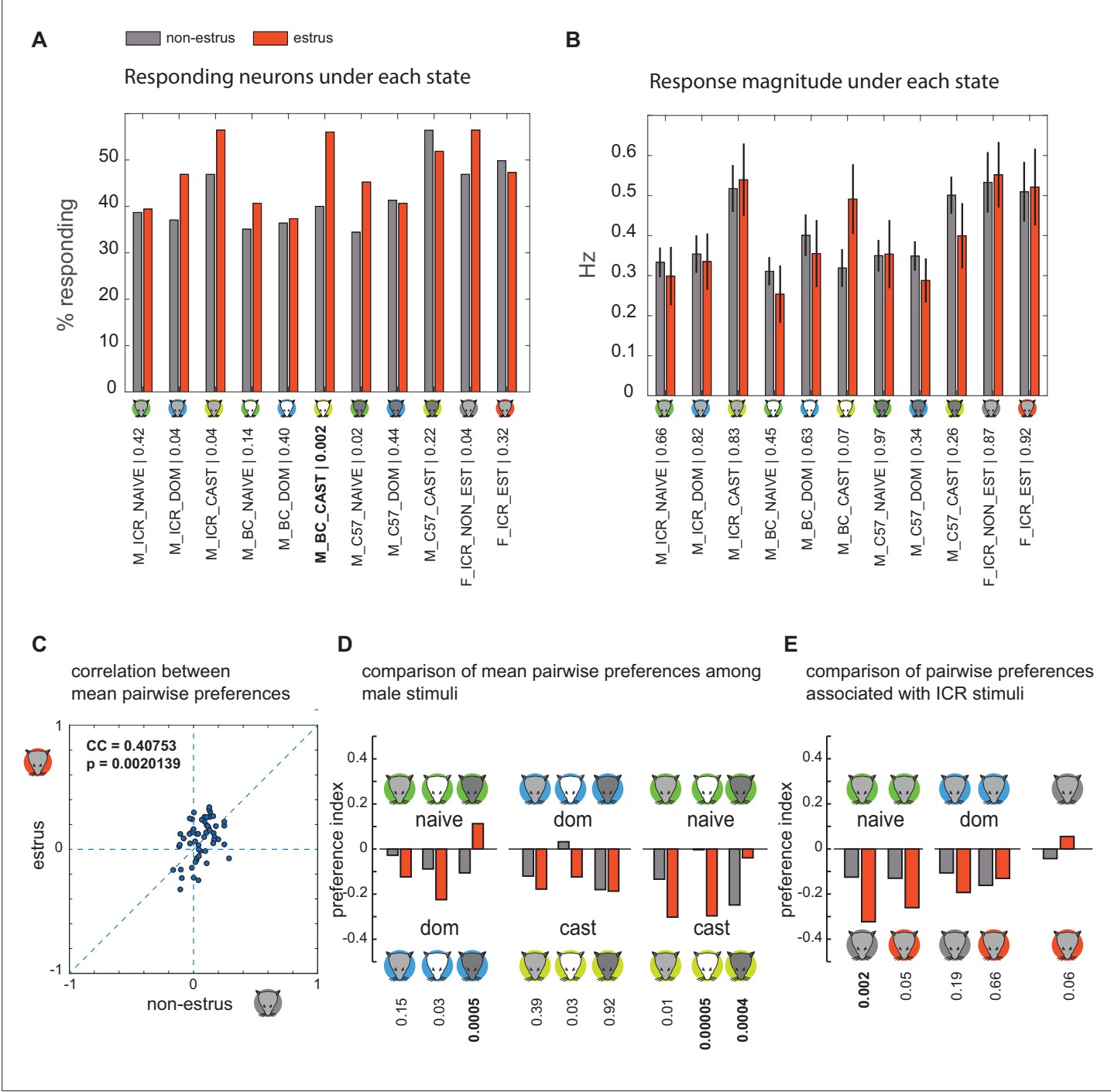

**Figure 5.** Comparison of responses in estrus and non-estrus females. (**A**) Percentage of responding neurons to each of the stimuli in non-estrus (gray) and estrus (red) neurons. p-Values correspond to the binomial exact test. (**B**) Mean response magnitude to each stimulus under the two states. p-Values correspond to nonparametric ANOVA. In both A and B, the Bonferroni-adjusted p-value given 11 comparisons, at the 0.05 level, is 0.0045. The single significant difference is indicated in bold. In A and B, n=305 non-estrus and n=241 estrus neurons. Error bars indicate the standard error of the mean. (**C**) Correlation between mean pairwise preference indices under the two reproductive states. The correlation coefficient and the probability to obtain it by chance are indicated. (**D**) Comparison of pairwise preference indices under the two states. Each pair of bars corresponds to one comparison. The stimuli are indicated by the icons and text. For example, the first pair of bars on the left corresponds to pairwise comparison of dominant and naive ICR male urine. In both estrus states, there is a stronger response to the dominant stimulus as indicated by the downward-pointing bars. However, the difference is not significant. (**E**) Comparison of pairwise preference indices between stimulus pairs comprising male and female stimuli. In D and E, significance is determined using a shuffling test (n=100,000). Bonferroni-adjusted p-values at the 0.05 level are 0.0056 and 0.01 for D and E, respectively. Significant differences among the states are indicated in bold. Note that in most cases, significant differences correspond to cases where responses during estrus are stronger to the less virile male stimulus.

associated with virility levels. To account for strain-related effects of response strength, we applied this analysis to each strain separately. Significant changes in preference are indicated by bold font in *Figure 5D* (nonparametric ANOVA, Bonferroni-corrected p-value for a nominal level of 0.05 is 0.0056). Significant changes include stronger responses to naive urine during estrus (C57), and two changes involving naive and castrated stimuli (BC, C57), yet in opposite directions. Summarizing, we find no consistent trends across the three strains, and specifically, no evidence for enhanced responses to virile stimuli during estrus.

A related possibility is that during estrus, responses to male stimuli will become stronger relative to female stimuli. To test this, we compared responses between female and (non-castrated) male stimuli from the ICR strain. In all cases (*Figure 5E*), female urine elicits stronger responses than male urine. The only significant difference (p=0.0024, Bonferroni-corrected p-value: 0.01) is that during estrus, responses to non-estrus female urine increase relative to naive male stimuli. Relative response strengths to estrus and non-estrus female urine also did not change during the estrus cycle (*Figure 5E*). In conclusion, relative response magnitudes are broadly conserved across estrus states. The changes that do take place do not reflect systematic modulation of sensitivity to particular virility states.

## Increased selectivity during estrus, but no consistent changes associated with virility state

Thus far, we examined hypotheses related to response magnitudes, reasoning that changes in physiological state can enhance responses to selected stimuli. Yet, another possibility is that individual neurons will become more *selective* to particular stimuli, without necessarily altering response magnitudes across the population. We first address this issue by examining lifetime sparseness, a normalized measure of selectivity based on response strength, ranging between 0 (uniform response magnitude) and 1 (exclusive response to one stimulus). As shown in *Figure 6A*, the sparseness of the population is higher for estrus neurons (p=0.0079, Kruskal-Wallis nonparametric ANOVA). To determine if these changes reflect selectivity to particular virility states, we examined them directly. Again, to account for potential strain-specific effects, we conducted this analysis for each strain separately. In the triangle plots (*Bergan et al., 2014*) of *Figure 6B*, vertices correspond to virility states, and dots represent the *relative* response magnitude of individual neurons to each virility state. Thus, dots near the center represent nonselective neurons, while selective neurons are represented by dots near vertices. The plots show that across all strains, dots corresponding to estrus and non-estrus neurons overlap widely.

To determine if selectivity changes with reproductive state, we calculated a response selectivity score (*Bergan et al., 2014*) for each neuron for each strain. The selectivity score ranges between 0 (uniform response strength) and 1 (exclusive response to one stimulus). We had hypothesized that during estrus, selectivity among male stimuli will be enhanced. This hypothesis appears true for ICR male stimuli, with a small yet significant increase in selectivity during estrus (p=0.0017, Kruskal-Wallis test). However, this is not the case for BC and C57-derived stimuli (*Figure 6C*) and is thus not a general feature of the estrus state.

Next, we analyzed pairwise response selectivity by considering, for each neuron, the *absolute* value of the preference index (see Materials and methods), which ranges between 0 (equal responses to both stimuli) and 1 (response to only one of the stimuli). We then compared the distributions of these indices across the two estrus states. The analysis (*Table 2*) is also not consistent with increased acuity for reproductively relevant cues during estrus. For example, of all pairwise comparisons involving a dominant and nondominant stimulus donor from the same strain, the only significant change involves a *decrease* in selectivity among dominant and naive C57 male urine during estrus. The pairwise comparisons (*Table 2*) also imply that the observed changes for the three-way comparisons for ICR stimuli mostly result from increased selectivity to castrated urine. In summary, while as a population, AOB-MCs from estrus females show somewhat sparser responses, there are no consistent shifts in selectivity for male virility state.

## Neuronal receptive field distributions across the estrus cycle

Finally, we return to examine response patterns and compare their frequencies across the estrus cycle. Such changes, if exist, could reflect shifts in tuning properties that would not necessarily be revealed by our previous analyses. Response pattern frequencies for units recorded in estrus and non-estrus females for the basic and adjusted patterns are shown in *Figure 7A and B*. Applying a binomial

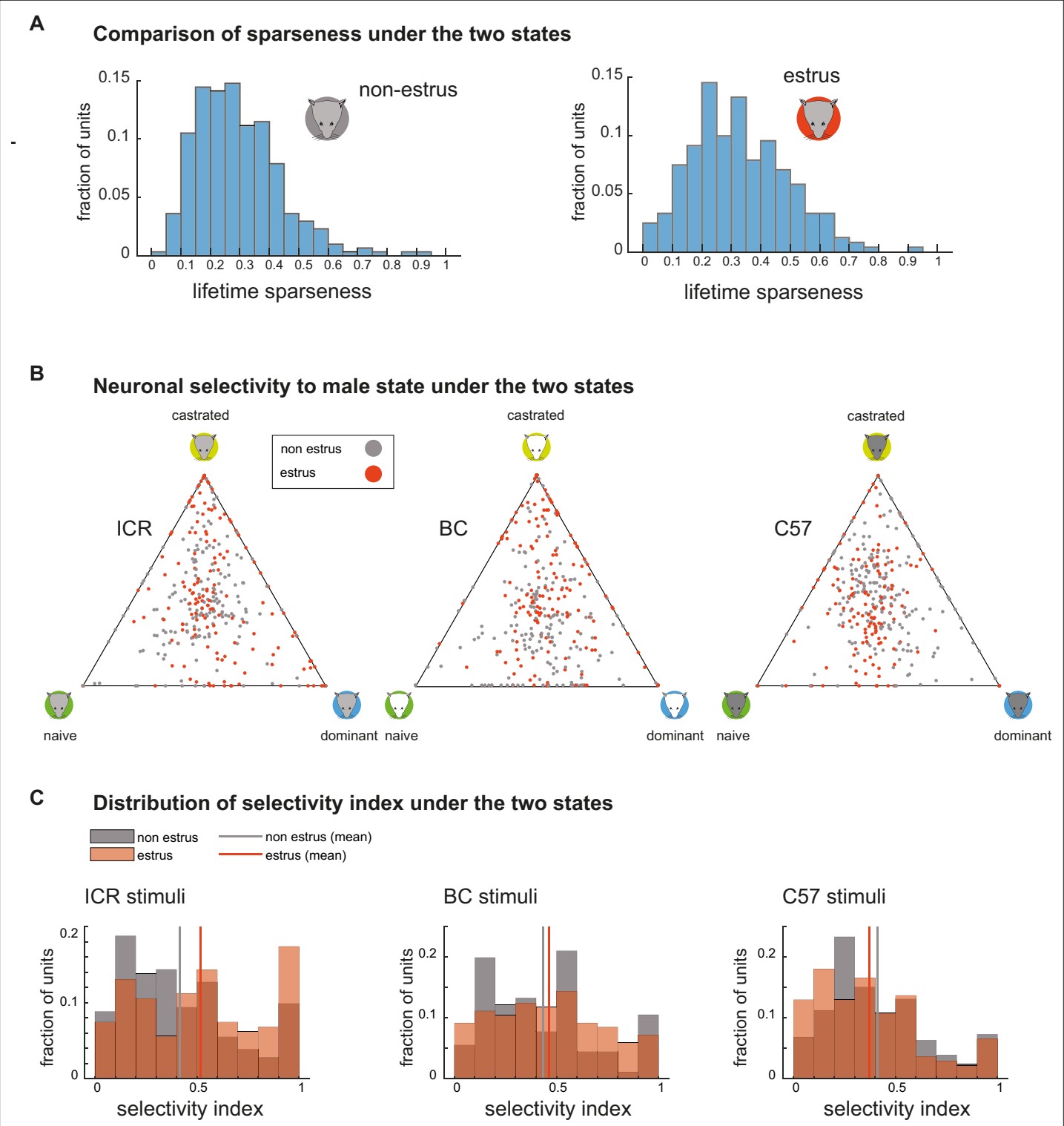

**Figure 6.** Response selectivity under the two reproductive states. (**A**) Lifetime sparseness distributions for non-estrus (left, n=305, mean: 0.29 median: 0.27) and estrus (right, n=241, mean: 0.33, median: 0.31) neurons. Nonparametric ANOVA p-value: 0.0078, reflecting higher selectivity during estrus. (**B**) Triangle plots showing relative response magnitude to stimuli from each of the three male states, separated according to strains. Neurons recorded in non-estrus and estrus females are indicated in gray and red, respectively. (**C**) Distributions of the male state selectivity index (calculated from the data shown in B) under the two reproductive states, calculated for each strain separately. Higher selectivity during estrus is observed for the stimuli from ICR male mice. p-Values corresponding to nonparametric analysis of variance (Kruskal-Wallis test): ICR: 0.0017, BC: 0.24, C57: 0.08 Mean selectivity scores: ICR: estrus: 0.52, n=161, non-estrus: 0.42, n=181. BC: estrus: 0.47, n=153, non-estrus: 0.43, n=181. C57: estrus: 0.37, n=139, non-estrus: 0.41, n=206.

**Table 2.** Detailed comparison of selectivity for pairs of stimuli between estrus and non-estrus neurons (related to *Figure 6*).

| Type of comparison | Stimulus 1 | Stimulus 2 | Non-estrus selectivity | Estrus selectivity | Selectivity score p-value |
|---|---|---|---|---|---|
| Female vs. female | F ICR estrus | F ICR non-estrus | **0.53788 (169)** | 0.49684 (142) | 0.34024 |
| Male vs. female | ICR dom | ICR estrus | 0.71497 (175) | **0.74204 (141)** | 0.99068 |
| | ICR dom | ICR non-estrus | 0.73335 (171) | **0.75566 (156)** | 0.82341 |
| | ICR naive | ICR estrus | 0.72123 (175) | **0.73349 (136)** | 0.81005 |
| | ICR naive | ICR non-estrus | 0.69395 (166) | **0.74774 (147)** | 0.26472 |
| Different states for a given strain | ICR dom | ICR naive | **0.60467 (138)** | 0.57776 (115) | 0.43316 |
| | C57 dom | C57 naive | **0.66205 (150)** | 0.46707 (110) | 3.9015e-05 |
| | BC dom | BC naive | **0.62197 (137)** | 0.56784 (103) | 0.25603 |
| | ICR dom | ICR cast | 0.67063 (166) | **0.73886 (154)** | 0.33714 |
| | C57 dom | C57 cast | **0.71072 (194)** | 0.6357 (127) | 0.064372 |
| | BC dom | BC cast | **0.73796 (155)** | 0.72499 (147) | 0.63383 |
| | ICR naive | ICR cast | 0.66641 (159) | **0.71219 (144)** | 0.3776 |
| | C57 naive | C57 cast | **0.6886 (180)** | 0.59994 (132) | 0.02337 |
| | BC naive | BC cast | **0.76303 (155)** | 0.75291 (145) | 0.49453 |
| Different strains for a given virility state | C57 naive | BC naive | **0.63828 (131)** | 0.62625 (114) | 0.60412 |
| | C57 naive | ICR naive | **0.64574 (139)** | 0.60728 (117) | 0.47299 |
| | BC naive | ICR naive | **0.60711 (134)** | 0.53642 (101) | 0.089909 |
| | C57 dom | BC dom | **0.68609 (152)** | 0.62328 (108) | 0.096686 |
| | C57 dom | ICR dom | **0.71036 (157)** | 0.62786 (118) | 0.039544 |
| | BC dom | ICR dom | **0.66277 (147)** | 0.56956 (114) | 0.036236 |
| | ICR cast | BC cast | **0.70931 (169)** | 0.58111 (157) | 0.00097307 |
| | ICR cast | C57 cast | **0.67162 (199)** | 0.60862 (147) | 0.10409 |
| | BC cast | C57 cast | **0.71471 (190)** | 0.58251 (144) | 0.0009967 |

p-Values are calculated using a nonparametric ANOVA (Kruskal-Wallis). Significant differences after adjustment for 23 comparisons (0.05/23=0.0022) are shown in red. Significant differences without adjustment (at the 0.05) are shown in blue. For each stimulus pair, values for the state with the higher selectivity are shown in bold font. The number of neurons in each comparison is indicated in parenthesis in the corresponding selectivity columns.

exact test (see Materials and methods), and correcting for multiple comparisons, none of the patterns show a significant state-dependent difference in frequency. After relaxing significance criteria (by removing the multiple comparisons correction), three differences emerge (bold labels in *Figure 7*). Again, however, none of these changes suggest an obvious ethological function, and, in particular, no changes in representation of dominance. In summary, response pattern distributions remain stable during the cycle, and the few changes that do occur are not compatible with enhanced selectivity to reproductively relevant stimuli.

## Discussion

The first stage in sexual behavior is the appetitive stage, which is further divided into detection, approach, and investigation (*Yin and Lin, 2023*). In rodents, the VNS is probably most important during investigation, which takes place after contact is established, allowing a thorough analysis of a potential partner's state (*Keller et al., 2009*; *Keller et al., 2006*). We had originally hypothesized that during estrus, responses to 'virile stimuli' will be accentuated, reflecting heightened sensitivity or discriminatory ability. Instead, we found that overall, AOB representations remain stable. This conclusion is based on analysis of response magnitudes to individual stimuli and relationships between

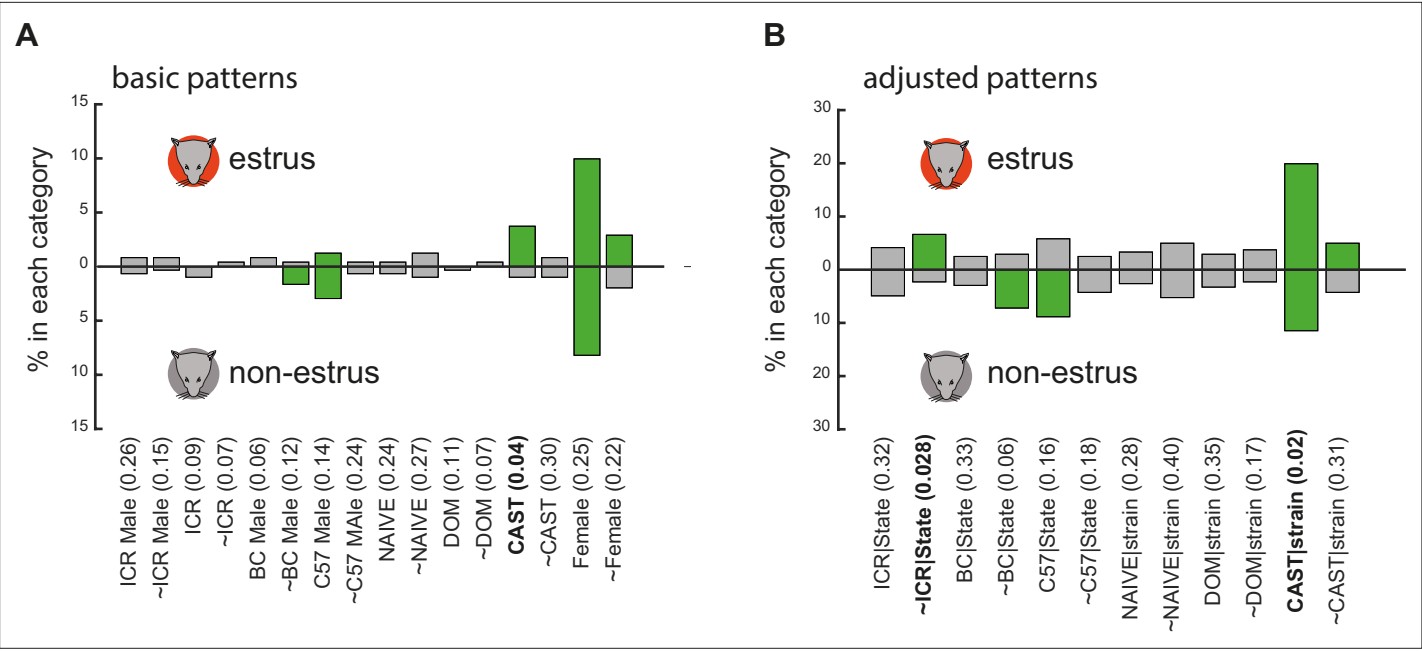

**Figure 7.** Comparison of response frequency under the two reproductive states. (**A**) Frequencies of basic patterns (including complementary patterns). (**B**) Frequencies of adjusted patterns (including complementary patterns). In both panels, frequencies in estrus and non-estrus females are shown above and below the horizontal axis, respectively. Significantly represented categories (using a shuffling test as described in *Figure 3*) are indicated in green. p-Values for differences for each pattern under the two states, using the binomial exact test, are listed. The Bonferroni-adjusted p-values at the 0.05 level are 0.0031 (0.05/16) and 0.0042 (0.05/12), for A and B, respectively, and thus none of the differences are significant. Relaxing the correction, three categories fulfill the 0.05 criterion (indicated in bold). None of them correspond to increased representation of dominant male stimuli during estrus. Note that in some cases, a category is overrepresented only during one of the states, but this is not necessarily reflected by a significant difference among the two states (e.g. ~female).

response magnitudes (*Figure 5*), selectivity to specific stimuli (*Figure 6*), population-level representations (*Figure 4*), and the frequency of response categories (*Figure 7*). It is consistent with our recent observations showing representational stability in virgin and in pregnant female mice, at least under temporal scales reflecting brief investigation bouts (*Yoles-Frenkel et al., 2022*). Combined with analysis of responses in male and female subjects (*Ben-Shaul et al., 2010*; *Bergan et al., 2014*) and in males from different strains (*Bansal et al., 2021*), we propose that the AOB conveys detailed information but does not 'interpret' it as a function of an organism's current state. Thus, the AOB provides an unbiased account of a potential partner's features, leaving evaluation of the appropriate response to downstream processing stages. Below, we first discuss what our study can reveal about response patterns of AOB neurons and then proceed to discuss the implications for state-dependent processing.

## Response patterns of AOB-MCs

While the primary focus of this study was state-dependent processing, we first considered the broader topic of response properties of AOB-MCs. Chemosensory space cannot be easily parameterized like visual or auditory space, and presentation of all potential ligands (and their combinations) is clearly unfeasible. Thus, receptive field properties of central chemosensory neurons, including AOB-MCs, are poorly characterized. The selective responses to mice from different sex and strain combinations that we observed here are consistent with previous electrophysiological (*Luo et al., 2003*; *Ben-Shaul et al., 2010*; *Tolokh et al., 2013*; *Bansal et al., 2021*) and chemical analyses (*Nagel et al., 2024*; *Fu et al., 2015*; *Cheetham et al., 2009*; *Schwende et al., 1986*; *Albone and Shirley, 1984*; *Lin et al., 2005*; *Zhang et al., 2007*). Selectivity to castrated vs. intact males has also been shown before (*Yoles-Frenkel et al., 2022*), while selectivity to male dominance is consistent with chemical analyses (*Thoß et al., 2019*) and immediate-early-gene expression profiles in the AOB (*Veyrac et al., 2011*). Here, we asked if AOB-MC response patterns reflect behaviorally relevant features of conspecifics. We found that most response patterns (*Figure 3*) are not more frequent than expected by chance.

What can we learn from our analysis of response pattern frequency? Generally, pattern frequency is determined by the fraction of molecules (or their combinations) that are common to stimuli that share some feature and the degree to which AOB-MCs sample these shared molecules (or their combinations). This idea is illustrated in *Figure 8*, where for simplicity, we ignore the fact that molecule levels are graded rather than binary, that multiple AOB-MCs can sample the same molecule, that the same molecules can be sampled by multiple AOB-MCs, and that trait selective information is often conveyed by combinations rather than single molecules. While these are critical facts, ignoring them does not alter the essence of our explanation. *Figure 8A* represents three individuals that share a common trait (e.g. dominance), each of which emits a certain number of molecules, chosen randomly from a 'world' of molecules (depicted in a 30×30 grid). Some of the molecules are unique to each individual (shown in green), while others are related to the common trait and thus shared by all individuals (red). The individual-selective molecules may convey information related to individuality or to some other unrelated trait. The three rows in *Figure 8A* represent scenarios with varying ratios between trait-selective and individual-selective molecules. Each row shows the molecules associated with each individual, as well as their union (right column).

To identify a given trait, it is needed to detect trait-selective molecules, and this involves the second element that dictates response frequency, namely the effective sampling properties of neurons (of course, AOB-MCs do not directly sample molecules, but rather VSNs, which do). Assuming some noise in production and detection of these molecules, robust trait detection should involve sampling of multiple molecules. Given a trade-off between robustness of detection and redundant sampling (since resources are limited), there is likely an optimal number of molecules/neurons for the identification of any given trait. We hypothesize that tuning properties of the AOB-MC population are adapted to efficiently sample trait-selective molecules. For each trait, three scenarios are possible: non-biased sampling, over-sampling, and under-sampling of trait-selective molecules (*Figure 8B*). Combined with the proportions of trait-selective and non-trait-selective molecules, these sampling statistics determine the frequency of particular response patterns across the population.

Taking these considerations into account, we speculate that optimal sampling by the VNS (reflecting both VNS tuning properties and AOB-MCs connectivity) involves selective sampling of rarely represented trait-related molecules and selective avoidance of common redundant molecules. Such an organization may optimize information retrieval with limited resources and can account for our observation that most patterns, which we assume correspond to meaningful traits, are not overrepresented.

However, if sampling is optimal, why are some patterns more frequent than expected by chance? Considering male and female stimuli, there may be numerous molecules that are more prevalent in females than in males. We speculate that under-sampling can reduce, but not overcome this excess. While recent analyses indicate that only a small number of urinary molecules are unique to all males or females from different strains (*Nagel et al., 2024*), our pattern types do not depend on unique, but rather on increased or decreased concentrations associated with specific traits. Similar considerations may apply to the C57 male category, which was also overrepresented in our data. Notably, analysis of chemical levels indicates that VSN selectivity for both sex and strain is less prominent than expected based on urinary contents (*Nagel et al., 2024*). This supports our ideas that VNS encoding acts to reduce the redundancy present in molecular levels.

The overrepresentation of castrated stimuli is puzzling since castrated males are not natural stimuli. Yet, this may precisely be the reason for the overrepresentation. Namely, if castration gives rise to sweeping changes in molecular composition, then the (hypothetical) evolutionary process of under-sampling could not happen. This would account for the puzzling fraction of castrated urine selective neurons and the strong responses to these stimuli (*Figure 2D and E*), observed here, and elsewhere (*Yoles-Frenkel et al., 2022*).

Along the same lines, in contrast to castrated stimuli, the dominant pattern was very infrequent (in fact, only one single unit in our dataset fulfilled it, *Figure 3C*). While this is not significantly less than expected by chance, it is surprising. We note that this is not because dominant stimuli are less potent than naive stimuli; they are generally stronger (see *Figure 2D and E*). Nor is it likely to be due to our modest sample size, as similar pattern distributions are observed when multiunit activity is also included in the analysis (not shown). Instead, we proposed that this scarcity may represent an optimal allocation of resources to separate dominant from naive males. Note also that population activity levels elicited by dominant and naive stimuli are similar (see *Figure 2B*), suggesting that

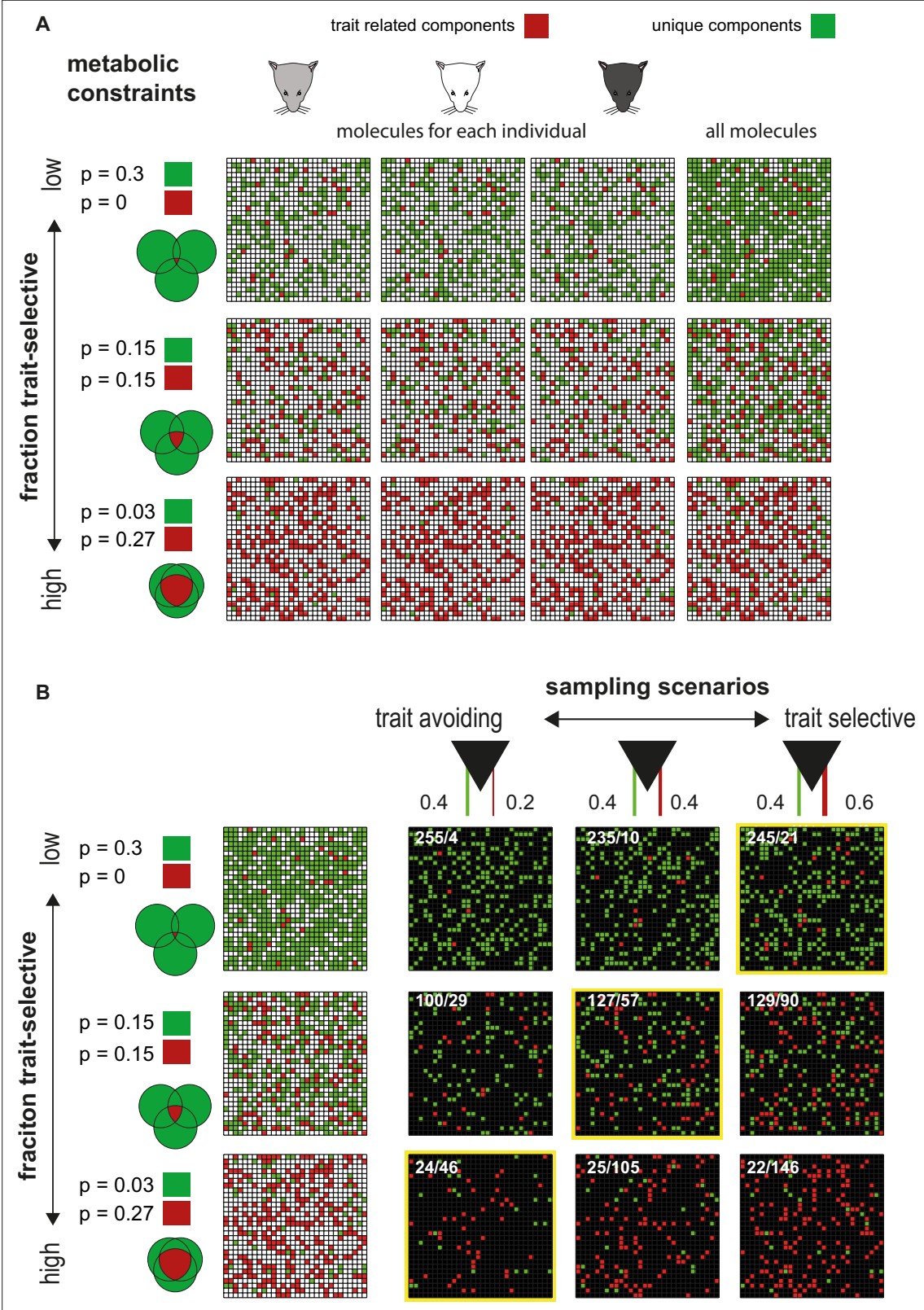

**Figure 8.** Schematic illustration of sampling of trait-selective molecules. (**A**) Molecular profiles associated with three individuals sharing a common trait. Trait-selective molecules are shown in red and all others (unique molecules) are shown in green, within a 30×30 grid representing all (900) molecules in the world. The three individuals are indicated on the three columns on the left. The rightmost column shows their union. Rows correspond to different proportions of individual specific and trait-selective molecules. (**B**) Different sampling scenarios by accessory olfactory bulb (AOB) neurons. The three

*Figure 8 continued on next page*

*Figure 8 continued*

rows correspond to the scenarios shown in A. The left column is the union of all molecules under each scenario, as shown in A. The three other columns represent sampling schemes that differ in the proportion of trait selective vs. other molecules sampled, moving from trait avoiding, to balanced, to trait selective sampling. The two values above each column indicate the probability of sampling unique (left) or common (trait-selective) molecules (right). We speculate that sampling properties are matched to component frequencies, thereby balancing robustness of trait detection with redundancy. We note that receptive fields of AOB-MCs are actually determined by vomeronasal sensory neurons (VSNs) and by their interactions with AOB-MCs.

unlike castration, dominance is not associated with extensive chemical changes. The number of strain-adjusted dominant patterns is not negligible (*Figure 3D*), implying that evaluation of male state might be done in the context of that male's strain. Admittedly, lacking knowledge of the chemical space associated with each of the stimuli, this and all the other ideas developed here remain speculative. A comprehensive analysis of tuning properties of AOB-MCs must take into account all the features that we ignored in our simple explanation and consider the balance between encoding multiple traits in parallel, some of which must involve overlapping molecules.

## Implications for female reproductive state-dependent processing

Considering the second theme of this research, namely state dependence, our results are not at odds with multiple reports of estrus-state-dependent behavioral changes in female mice. They are also reconcilable with state-dependent representations in downstream regions that reflect fluctuating hormonal states. However, our findings are unexpected in light of estrus-state-dependent changes in the earlier processing stage, namely at the level of VSNs (*Dey et al., 2015*; *Cherian et al., 2014*; *Eckstein et al., 2020*). One possible explanation is that our approach cannot detect finer adaptations within dedicated processing streams (*Ishii et al., 2017*). This scenario applies in *Dey et al., 2015*, which showed changes in a subset of VSNs sensitive to male stimuli. Two other studies revealed more global sex hormone-dependent changes in VSNs (*Cherian et al., 2014*; *Eckstein et al., 2020*). Lacking knowledge of the functional implications of such changes on the AOB network, it is difficult to explain why we did not observe broad changes. One simple possibility is that such effects will generally increase excitability. However, we did not observe such changes, perhaps due to compensatory mechanisms within the AOB. Finally, we note that despite the global stability, some of our comparisons yielded significant differences among the two states. While these differences do not represent increased (or decreased) responsiveness to virile stimuli, we cannot rule out their connection with the changes reported in VSNs.

Our experimental approach has inherent advantages and disadvantages for detecting state-dependent effects. Since mice are anesthetized, changes involving active exploration or arousal cannot be detected. This includes the possible effects of feedback that may themselves be state-dependent, such as those from estrogen receptor-expressing neurons in the medial amygdala (*Inbar et al., 2022*). On the other hand, in awake mice, there is no simple way to monitor the dynamics of stimulus uptake, which may vary with state, and thus confound state-dependent neuronal responses with state-dependent sampling. The anesthetized preparation ensures that stimulus uptake is consistent under both states and allows repeated delivery of multiple stimuli, regardless of the subject's engagement or motivation. In this context, we note that MOS activation using our preparation is likely minimal, thereby reducing its influence on sampling behavior, or via more direct forms of cross talk between the two systems.

In conclusion, the two themes of this study inspire two future research directions. The first calls for continued investigation of state-dependent representations, in both the AOB under the awake state and in deeper brain regions, in both anesthetized and ultimately awake animals. The second direction involves a broader effort to understand how behaviorally relevant features are encoded in chemical space, in early sensory processing, and eventually, decoded by downstream neurons that receive sensory inputs.

## Materials and methods

### Subjects

For recordings, we used sexually inexperienced adult female mice from the ICR strain (Envigo Laboratories, Israel). All experiments complied with the Hebrew University Animal Care and Use Committee (protocol number MD-13-13715-3).

### Determination of the estrus stage

Female estrus stage was determined using a vaginal lavage of 15 µl of saline. The saline sample was then deposited on a Superfrost Plus slide (Thermo Scientific) and dried. Slides were treated with Wright Stain Solution (Sigma) and photographed under a light microscope. The estrus stage was determined using the cytological composition of the smear, as described in *Byers et al., 2012*; *Caligioni, 2009*.

### Male status

Our stimulus set includes urine samples from castrated, naive, and dominant proven breeder mice. Proven breeder and naive male mice were purchased (Envigo Laboratories, Israel) and then assigned to cages containing two proven breeders and two to three naive mice. One to two days after mice were placed in these cages, we identified candidate dominant male (always one of the two proven breeders) by observation of their home cages. To confirm dominance, we conducted dominance tests involving each proven breeder mouse with each of the other cage mates (including all naive cage mates and the other proven breeder male). The test included a 'tube test', conducted in a new cage with clean bedding. The two mice were placed at opposite ends of a cylindrical tube, and the mouse that forced the other to exit was designated as dominant. In addition, immediately following this tube test, the same pair of mice was monitored for an additional 10 min, and the mouse displaying aggressive behavior (biting, chasing, tail rattling) toward the other was identified as dominant. We only classified mice as dominant if they showed dominance during the tube test and the subsequent observation period in the cage. We also confirmed that the established hierarchy was maintained throughout the entire period of urine collection, over the course of ~4 weeks, by monitoring the behavior of these mice in their home cages. Dominant male mice consistently showed aggressive behaviors, whereas non-dominant mice engaged in self-grooming behaviors more frequently. Castrated male urine was collected from 8-week-old males that had their testes removed under isoflurane anesthesia at the age of 4 weeks (before puberty).

### Stimuli and dataset

Stimuli were collected from castrated, naive, and sexually experienced male mice from the BALB/C, C57BL/6, and ICR strains and estrus and non-estrus sexually naive adult female mice from the ICR strain (Envigo Laboratories, Israel). For urine collection, mice were gently held over a plastic sheet until they urinated. The urine was collected in plastic tubes, flash-frozen in liquid nitrogen, and then stored at –80°C. Each of the samples presented here comprised urine from three different individuals, diluted 1:10 in Ringer's solution. Urine from each individual was collected over a period of ~4 weeks and pooled before combining it with urine from males of the same strain and status.

### Surgical procedures and electrode positioning

Experimental procedures were detailed in *Yoles-Frenkel et al., 2017*. Briefly, mice were anesthetized with 100 mg/kg ketamine and 10 mg/kg xylazine, tracheotomized, and a cuff electrode was implanted around the sympathetic nerve trunk. Incisions were closed with Vetbond (3 M) glue, and the mouse was placed in a stereotaxic apparatus where anesthesia was maintained throughout the experiment (0.5–1% isoflurane in oxygen). A craniotomy was made immediately rostral to the rhinal sinus, the dura was removed, and electrodes were advanced into the AOB at an angle of ~30° with an electronic micromanipulator (MP-285; Sutter Instruments, Novato, CA, USA). We used 32-channel probes (NeuroNexus Technologies, Ann Arbor, MI, USA) with 8 channels on each of four shanks, with the following specification: (horizontal) distance between the shanks: 200 µm, site area: 177 µm$^2$, and (vertical) within-shank site spacing either 50 µm or 100 µm (NeuroNexus acute probe models: A4x8-5 mm-100-200-177-A32 and A4x8-5 mm-50-200-177-A32). Before recordings, electrodes were

dipped in fluorescent dye (DiI, Invitrogen, Carlsbad, CA, USA) to allow subsequent confirmation of placement within the AOB external cellular layer.

## Stimulus presentation procedure

Each session contains several distinct stimuli. Typically, each stimulus was presented repeatedly in a pseudorandom order, at least four times, and typically five times within each session across all sessions (number of repeated presentations per stimulus: 5.1±0.9, mean ± sd. In 72% of the cases, stimuli were given five or more times. Otherwise, they were presented four times). Stimuli were presented in blocks, and within each block, all stimuli were presented in a random order. An ITI of 18 s was applied between stimulus presentations. During each presentation, 2 µl of stimulus was manually applied directly into the nostril using a micropipette (*Yoles-Frenkel et al., 2017*). After a delay of 20 s, a square-wave stimulation train (duration: 1.6 s, current: ±120 µA, frequency: 30 Hz) was delivered through the sympathetic nerve cuff electrode to induce vomeronasal organ suction. Following another 40 s delay, the nasal cavity was flushed with 1–2 ml of Ringer's solution and drained with suction from the nasopalatine duct. The cleansing procedure lasted 50 s and included sympathetic trunk stimulation to facilitate stimulus elimination from the vomeronasal organ lumen.

## Electrophysiology

Neuronal data was recorded using an RZ2 processor, PZ2 preamplifier, and two RA16CH head-stage amplifiers (TDT, Alachua, FL, USA). Single-unit and multiunit activity were sampled at 24,414 Hz and band-pass filtered at 300–5000 Hz. Custom MATLAB codes were used to extract spike waveforms. Spikes were sorted automatically according to their projections on two principal components on eight channels of each shank using KlustaKwik (*Csicsvari et al., 2003*; *Hazan et al., 2006*) and then manually verified and adjusted using Klusters (*Hazan et al., 2006*). Spike clusters were evaluated by their waveforms, projections on principal component space (calculated for each session individually), and autocorrelation functions. A cluster was defined as a single unit if it displayed a distinct spike shape, was fully separable from both the origin (noise) and other clusters, and if its autocorrelation function demonstrated a clear trough around time 0 of at least 10 ms. Clusters apparently comprising more than one single unit were designated as multi-units. Under these definitions, multiunits could represent the activity of as few as two units. Such units are not included in our dataset. As a further criterion for single-unit classification, we examined the 'shape symmetry' of all spike shapes. Shape symmetry was defined as the correlation coefficient of the two sides (margins) of the spike waveform around its peak (in absolute value, with one of the margins reversed). For example, if the waveform contains nine samples with a peak at sample 5, the correlation will be calculated between the vectors defined by samples 1–4, and a reversed version of the vector defined by samples 6–9. If the shape is symmetric, the symmetry will be 1. When the peak is not in the middle, as is usually the case, the correlation is calculated for vectors determined by the smaller margin. A high symmetry (>0.95) is helpful for identifying noise accidentally misclassified as spikes. Using a graphical representation of all spike shapes, we examined the entire dataset for such noise-like spikes and removed them from further analyses.

## Data analysis

All data analyses and visualizations were performed using custom and built-in MATLAB code. The response of a unit to a given stimulus was defined as the average firing rate change over a 60 s window following stimulus application (change measured compared to the 17 s period preceding application). This broad response window covers both the post-application and post-stimulation periods, as in *Bansal et al., 2021*. This window was chosen because in some cases, responses began after stimulus application and prior to electric stimulation of the SNT. Response significance of a given unit to a given stimulus was determined by comparing its spiking rate distribution following all repeats to the baseline firing frequency distribution during the 17 s period prior to stimulus application. Response significance (of a particular unit to a given stimulus) was determined by a nonparametric analysis of variance (Kruskal-Wallis test) comparing the set of poststimulation rates to the set of preceding baseline rates (preceding rates were pooled across all stimuli), using a threshold of 0.05. In the image in *Figure 2A* (and in other related representations), the response magnitude of each unit was normalized by the absolute value of the maximal response (i.e. the response to the stimulus

that elicited the strongest response, in absolute value). Thus, all response values range between –1 and 1. In addition, in this image and in some analyses, all nonsignificant responses were set to 0. For *Figure 2B*, we applied classical multidimensional scaling, using the data in each of the rows in *Figure 2A*, using the MATLAB *cmdscale* function, and the *correlation* distance metric (defined as one minus the sample correlation).

In a preliminary stage of analysis, we sought to define response patterns according to response significance (e.g. with a dominant pattern implying that all dominant stimuli elicit significant responses, while all other stimuli do not). However, this approach yielded a very small number of neurons conforming to each pattern. Thus, we ultimately applied more permissive criteria. These criteria rely on relative response magnitudes, as shown in *Figure 3A*. For example, an ICR male pattern (upper row in the left matrix in *Figure 3A*) is fulfilled by neurons whose responses to each of the three ICR male stimuli are *stronger* than responses to all of the other stimuli (regardless of response significance). In the complementary pattern (~ICR male, shown in the upper row in central matrix), the response to each of these stimuli must be *smaller* than to all the other stimuli. As one example of a strain-adjusted pattern, to qualify for a dominant/strain pattern, a neuron's response to the dominant stimulus must be stronger than responses to all other male stimuli from the same strain. In other words, strain (or state) adjusted patterns involve three distinct conditions, as shown in the right panel of *Figure 3A*. Note that the adjusted patterns involve more permissive criteria than the non-adjusted patterns. Thus, for example, a dominant pattern is by definition also a strain-adjusted dominant pattern (dominant/ strain).

To analyze pattern frequencies, for each pattern, we first found the number of neurons that conform to each of the patterns. Then, we calculated the probability to observe that number of cases by 'chance'. To analyze if response frequencies occur more often than expected by chance, we constructed a null distribution by shuffling the response data matrix. In this matrix (as in *Figure 2A*), each row corresponds to one neuron, and each column corresponds to one stimulus. In each of 100,000 iterations, we randomly shuffled the order of all elements within a column and calculated the frequency of each response pattern. This frequency for each pattern was calculated across all iterations, allowing us to evaluate the probability to obtain the observed frequencies given only the marginal probabilities to each stimulus.

To correlate global response similarity between neurons recorded in estrus and non-estrus female, we used the MATLAB *pdist* function, which calculates pairwise distances along arrays. The arrays in our cases were the population responses (after normalization, as shown in the matrices in *Figure 4A*). We used the *correlation* and *Euclidean* distance measures for *Figure 4B, D* and *Figure 4C, E* respectively. We used the MATLAB *corrcoef* function to calculate the correlation coefficient and the p-value (the probability to observe such a large correlation under the hypothesis of no correlation).

To calculate the significance of differences in the number of neurons responding under each of the two states (*Figure 5A*), we tested the null hypothesis that response probability is identical under both states. This was done by calculating the observed probability (based on the total number of responding neurons, per stimulus, across both states) and then calculating four different probabilities under the binomial distribution: the probability to obtain the observed number or more responding neurons in estrus, the probability to obtain the observed number or less of responding neurons in estrus, and the corresponding probabilities for neurons recorded in non-estrus females. The smallest (i.e. most unlikely) value of these four probabilities was then considered as the p-value. Since the smallest p-value is taken, this yields a sensitive test of differences. To calculate the significance of differences between mean response magnitudes for each stimulus (*Figure 5B*), we used a two-tailed nonparametric analysis of variance (Kruskal-Wallis test), comparing the distributions of response strengths across all neurons recorded in estrus and in non-estrus females. A Bonferroni correction for multiple corrections was applied. For two-way ANOVA, we used the MATLAB *anovan* function, with stim identity and female state as factors, using a *full* model.

The pairwise preference index $PI_{AB}$ for two stimuli A and B (as shown in *Figure 5C*) is defined as $(R_A–R_B)/(R_A + R_B)$, where $R_A$ and $R_B$ are the responses to stimuli A and B after rectification (i.e. after setting negative responses to zeros). Rectification was applied to keep the range of the preference indices between –1 and 1. For each pair of stimuli, we only considered neurons that showed a significant response to at least one of the stimuli. Then, the mean preference indices were calculated for each stimulus pair, as shown in *Figure 5C*, for neurons recorded in estrus and non-estrus females. The

correlation coefficient and p-value were derived using the *corrcoef* MATLAB function, as described above.

To compare pairwise preference indices across states (*Figure 5D and E*), we also used a bootstrapping approach. Specifically, we first found the absolute difference between the mean preference indices for a given stimulus pair under the two states. Then, we repeatedly shuffled the dataset (effectively, neurons were randomly assigned to one of the two reproductive states) 100,000 times and calculated the absolute difference across the two (shuffled) states. Then, we derived the p-values as the fraction of cases in the shuffled distribution which were larger than the actual observed distribution. In other words, these p-values reflect the probability of obtaining such a state-dependent difference in preference indices by chance.

To calculate lifetime sparseness (S), we used the definition given in *Vinje and Gallant, 2000*:

$$S = \left[ 1 - \frac{\left( \sum_i |r_i| / N \right)^2}{\sum_i (r_i^2) / N} \right] / \left[ 1 - \frac{1}{N} \right]$$

where $r_i$ is the response of neuron to the ith stimulus (averaged across all trials) and N is the number of stimuli (N=11 in our case). S varies between 0 and 1, with 0 indicating uniform responses to all stimuli and 1 indicating a response to only one stimulus. Lifetime sparseness distributions across states were compared using a nonparametric analysis of variance (Kruskal-Wallis).

Triangle plots in *Figure 6B* were generated by considering the response of each neuron to each of three stimuli. Negative responses were truncated to 0 in this analysis. The position of each point is determined by the relative weights of three unit vectors pointing from the triangle center to each of the three vertices. The selectivity index (*Figure 6C*) was calculated for each neuron (and each stimulus triplet) as in *Bergan et al., 2014*. Specifically, it is the normalized distance from the center of the triangle, with values ranging between 0 (lowest selectivity, center of triangle) and 1 (highest selectivity, vertices of the triangles).

$$Si = \frac{\sqrt{(a-m)^2 + (b-m)^2 + (c-m)^2}}{d}$$

where a, b, and c are the responses to each of three stimuli. The value corresponding to equal responses to all stimuli is m, which is equal to 1/3. d is the maximum possible value of the numerator of the equation and is equal to $\sqrt{2/3}$. Selectivity index distributions under the two states were compared using a nonparametric analysis of variance (Kruskal-Wallis).

For comparison of category frequency distributions under the two states (*Figure 7A and B*), we applied a binomial exact test under the hypothesis of equal distributions under two states. Essentially, we followed the same procedure described above for comparing the number of responsive neurons, with the difference that here we calculated the probability to obtain a certain number of neurons corresponding to each pattern. Bars in *Figure 7* are shaded in green if the probability to obtain the observed pattern frequency (calculated separately for each state) is smaller than 0.05, employing the same shuffling approach described for *Figure 3C*.

## Acknowledgements

We thank Michal Yoles-Frenkel, Anat Kahan, and Karen Marom for help and support throughout this project. We thank Dr. Annika Cichy for helpful comments on the text. This work was supported by the Israeli Science Foundation (ISF) grant (1703/16).

## Additional information

### Funding

| Funder | Grant reference number | Author |
|---|---|---|
| Israel Science Foundation | 1703/16 | Yoram Ben-Shaul |

The funders had no role in study design, data collection and interpretation, or the decision to submit the work for publication.

### Author contributions

Oksana Cohen, Conceptualization, Investigation, Methodology; Yoram Ben-Shaul, Conceptualization, Formal analysis, Supervision, Funding acquisition, Writing – original draft, Writing – review and editing

### Author ORCIDs

Yoram Ben-Shaul  https://orcid.org/0000-0002-0407-4221

### Ethics

All experimental procedure described in this work complied with the Hebrew University Animal Care and Use Committee (protocol number MD-13-13715-3).

Reviewer #1 (Public review): https://doi.org/10.7554/eLife.103959.3.sa1
Author response https://doi.org/10.7554/eLife.103959.3.sa2

## Additional files

### Supplementary files

MDAR checklist

### Data availability

The code and data for all analyses are available at the public github repository: https://github.com/yorambenshaul/state-dependent-processing-cohen-ben-shaul (copy archived at *Ben-Shaul, 2025*). The repository includes a Matlab data file containing all the electrophysiological data required for the analyses, a set of matlab code files (m files), and instructions on how to generate the figures in this manuscript using the files.

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
