## [Editor Report · eLife Assessment]

This **important** work substantially advances our understanding of how accessory olfactory bulb neurons respond to social odor cues across the estrous cycle, showing that responses vary with the strain and sex of the odor source but display no consistent differences between estrous and non-estrous states. It employs a unique electrophysiology preparation that activates the vomeronasal organ pump via electric stimulation, enabling precise recordings of accessory olfactory bulb cell responses to different chemosignals in anesthetized mice. Overall, the study presents **convincing** findings on the stability and variability of accessory olfactory bulb response patterns, indicating that while accessory olfactory bulb detects social signals, it does not appear to interpret them based on reproductive state. This work will be of interest to those studying olfaction, social behavior, reproductive cycles, and systems neuroscience more broadly.

---

## [Referee Report · Reviewer #1 (Public review)]

Summary:

In this detailed study, Cohen and Ben-Shaul characterized the AOB cell responses to various conspecific urine samples in female mice across the estrous cycle. The authors found that AOB cell responses vary with strains and sexes of the samples. Between estrous and non-estrous females, no clear or consistent difference in responses was found. The cell response patterns, as measured by the distance between pairs of stimuli, are largely stable. When some changes do occur, they are not consistent across strains or male status. The authors concluded that AOB detects the signals without interpreting them. Overall, this study will provide useful information for scientists in the field of olfaction.

Strengths:

The study uses electrophysiological recording to characterize the responses of AOB cells to various urines in female mice. AOB recording is not trivial as it requires activation of VNO pump. The team uses a unique preparation to activate the VNO pump with electric stimulation, allowing them to record AOB cell responses to urines in anesthetized animals. The study comprehensively described the AOB cell responses to social stimuli and how the responses vary (or not) with features of the urine source and the reproductive state of the recording females. The dataset could be a valuable resource for scientists in the field of olfaction.

Weaknesses:

The study will be significantly strengthened by understanding the "distance" of chemical composition in different urine. This could be an important future direction.

---

## [Author Response]

The following is the authors’ response to the original reviews

**Public Reviews:**

**Reviewer #1 (Public review):**
Summary:In this detailed study, Cohen and Ben-Shaul characterized the AOB cell responses to various conspecific urine samples in female mice across the estrous cycle. The authors found that AOB cell responses vary with the strains and sexes of the samples. Between estrous and non-estrous females, no clear or consistent difference in responses was found. The cell response patterns, as measured by the distance between pairs of stimuli, are largely stable. When some changes do occur, they are not consistent across strains or male status. The authors concluded that AOB detects the signals without interpreting them. Overall, this study will provide useful information for scientists in the field of olfaction.Strengths:The study uses electrophysiological recording to characterize the responses of AOB cells to various urines in female mice. AOB recording is not trivial as it requires activation of VNO pump. The team uses a unique preparation to activate the VNO pump with electric stimulation, allowing them to record AOB cell responses to urines in anesthetized animals. The study comprehensively described the AOB cell responses to social stimuli and how the responses vary (or not) with features of the urine source and the reproductive state of the recording females. The dataset could be a valuable resource for scientists in the field of olfaction.Weaknesses:(1) The figures could be better labeled.

We revised all figures (except the model figure, Fig. 8), and among other improvements (many of which were suggested by the reviewers in other comments), added more labelling and annotation within the figures.

(2) For Figure 2E, please plot the error bar. Are there any statistics performed to compare the mean responses?

We added error bars (standard errors of the mean). We had not originally performed statistical comparisons between the stimuli, but now we have. The analysis of responses strength now appears in a new table (Table 1)

(3) For Figure 2D, it will be more informative to plot the percentage of responsive units.

Done.

(4) Could the similarity in response be explained by the similarity in urine composition? The study will be significantly strengthened by understanding the "distance" of chemical composition in different urine.

We agree. As we wrote in the Discussion: “Ultimately, lacking knowledge of the chemical space associated with each of the stimuli, this and all the other ideas developed here remain speculative.” We note however, that chemical distance (which in itself is hard to define) will provide only part of the picture. The other part is the “projection” of chemical space on the receptor array. This is an idea that we develop in the Discussion and in Figure 8. Specifically, that it is the combination of stimulus composition, and receptor tuning properties that will determine stimulus distances in neuronal space.

That said, a better understanding of the chemical distance is an important aspect that we are working to include in our future studies. For this dataset unfortunately, we have no such data.

(5) If it is not possible for the authors to obtain these data first-hand, published data on MUPs and chemicals found in these urines may provide some clues.

This comment is directly related to the previous one. Measurements about some classes of molecules may be found for some of the stimuli that we used here, but not for all. We are not aware of any single dataset that contains this information for any type of molecule across the entire stimulus set that we have used and pooling results from different studies has limited validity because of the biological and technical variability across studies. In order to reliably interpret our current recordings, it would be necessary to measure the urinary content of the very same samples that were used for stimulation. Unfortunately, we are not able to conduct this analysis at this stage.

(6) It is not very clear to me whether the female overrepresentation is because there are truly more AOB cells that respond to females than males or because there are only two female samples but 9 male samples.

The definitive answer to this comment is given in our response to the next one.

Nevertheless, we agree that this is an important point. It is true that the number of neurons fulfilling each of the patterns depends on the number of individual stimuli that define it (and on the frequency of neurons that respond to those stimuli). However, our measure of “over representation” was designed to overcome this bias, by using bootstrapping to reveal if the observed number of patterns is larger than expected by chance. The higher frequency of responses to female, as compared to male stimuli, is observed in other studies by others and by us, also when the number of male and female stimuli is matched (e.g., Bansal et al BMC Biol 2021, Ben-Shaul et al, PNAS 2010, Hendrickson et al, JNS, 2008). However, here, by overrepresentation, we do not refer to the higher frequency of female responding neurons, but rather that given the number of responding neurons, the *female* pattern is more common than expected by chance.

(7) If the authors only select two male samples, let's say ICR Naïve and ICR DOM, combine them with responses to two female samples, and do the same analysis as in Figure 3, will the female response still be overrepresented?

Following this suggestion, we have performed this analysis, and we were glad to see that the result is the one we had anticipated. Below, we provide an image of the results, following the same approach that we applied before, and showed in Figure 3C. Here, we defined a female pattern (using the two female samples) and compared it to a male pattern (using the ICR naïve and ICR DOM as suggested). It is as if we had only four stimuli in our set. As in the article, we calculated the expected distribution with 100,000 shuffles. We denoted this pattern as F/M ICR. The results are shown below.

Under the present conditions, the distribution of the number of female selective patterns is larger i.e., shifted to the right, compare to the female category in Figure 3C. This is expected, since now the criterion is more permissive. Specifically, now to qualify as a “female pattern”, the two responses to female urine must be stronger only than the responses to the two male stimuli included in this analysis (and to all other responses). Notably, although the null distribution shifted to the right, the actual number of neurons fulfilling this pattern is also larger, so that the actual number remains significantly larger than expected by chance. This is also true for the reverse category (as is the case in the ~female category Figure 3C). Thus, we conclude that overrepresentation of the female pattern is not a trivial consequence of the number of male and female stimuli.

**Author response image 1. sa2fig1:** 

(8) In Figure 4B and 4C, the pairwise distance during non-estrus is generally higher than that during estrus, although they are highly correlated. Does it mean that the cells respond to different urines more distinctively during diestrus than in estrus?

This is an important observation (!) and we had originally overlooked it. It is true that higher distance (as they are in estrus) imply more distinct population level responses and hence better discrimination among stimuli. However, this is inconsistent with all our other analyses that do not point to enhanced selectivity or discrimination in either state. If anything, we find somewhat higher sparseness in estrus. Yet, there may be technical explanations for the differences.

For Euclidean distances, the explanation may be trivial. The distance depends on the number of dimensions (i.e., units), and since our sample contains more neurons recorded during non-estrus, the larger distance is expected.

In fact, there is a similar dependence on sample size for the correlation distance. Smaller samples are associated with higher (spurious) correlations, and hence larger samples are be associated with larger distances. To demonstrate this, we conducted a simple simulation, where we calculated the absolute correlation coefficients of random samples from standard normal distributions (using the MATLAB function randn), changing the size of the population. For each sample size, we conducted 1000 tests. We considered sample sizes from 10 to 100000, including 200 and 300 (which are similar to our sample sizes). The results are shown in the figure below. Note that the absolute value of the correlation coefficient decreases with sample size, while the p-value for the observed correlation is stable at ~0.5.

While this is not a rigorous analysis of this issue, and while it does not exactly reflect the scenario in our data, where correlations are generally positive, it shows that the observed correlation (and hence correlation distance) is also affected by sample size.

For these reasons, we focus on comparison of these distances, rather than the absolute values of the correlation distances.

Following this comment, we now write in the manuscript:

“We first note that distances are generally larger during non-estrus, suggesting enhanced discrimination during this stage. However, further analyses of sparseness and selectivity do not support this idea (see below). Furthermore, we note that both Euclidean and correlation distances generally depend on sample size. In both cases, distances are expected to increase as a function of sample size, which in our dataset, is larger for the non-estrus (n = 305) as compared to the estrus (n = 241) neurons. Because of this factor, we focus here on the similarity of the relative within-state distances across the states (and not on their absolute magnitudes). Specifically, we find a positive and significant correlation among pairwise population distances under the two states. Thus, at the population level, representational space remains broadly stable across the estrus cycle. Nevertheless, several points in Fig. 4D, E clearly diverge from a linear relationship, implying that representational space differs under the two states. We next examine such state-dependent changes in more detail.”

(9) The correlation analysis is not entirely intuitive when just looking at the figures. Some sample heatmaps showing the response differences between estrous states will be helpful.

If we understand correctly, the idea is to show the correlation matrices from which the values in 4B and 4C are taken. The relevant images are now included in figure 4B, C and are references within the main text.

**Reviewer #2 (Public review):**
Summary:Many aspects of the study are carefully done, and in the grand scheme this is a solid contribution. I have no "big-picture" concerns about the approach or methodology. However, in numerous places the manuscript is unnecessarily vague, ambiguous, or confusing. Tightening up the presentation will magnify their impact.

We have reviewed the text and made substantial editing changes. Along with other specific comments by made both reviewers, we hope that these changes improve the presentation.

Strengths:(1) The study includes urine donors from males of three strains each with three social states, as well as females in two states. This diversity significantly enhances their ability to interpret their results.(2) Several distinct analyses are used to explore the question of whether AOB MCs are biased towards specific states or different between estrus and non-estrus females. The results of these different analyses are self-reinforcing about the main conclusions of the study.(3) The presentation maintains a neutral perspective throughout while touching on topics of widespread interest.Weaknesses:(1) Introduction:The discussion of the role of the VNS and preferences for different male stimuli should perhaps include Wysocki and Lepri 1991

We assume that the reviewer is referring to “Consequences of removing the vomeronasal organ” by Wysocki CJ, Lepri JJ, a review article in J Steroid Biochem from 1991. We were not familiar with this specific article and have now read it. The article discusses various male behaviors, and some effects on female behavior and physiology (e.g., puberty acceleration, maternal behaviors, ovulation) but we could not find any mention of the preference of female mice in this article. We also expanded our search to all pubmed articles authored by Wysocki and Lepri and then all articles by Wysocki (with the keyword Vomeronasal). Despite our best intentions to give due credit, we found nothing that seems directly related to this statement. Please correct us if we had missed anything.

(2) Results:a) Given the 20s gap between them, the distinction between sample application and sympathetic nerve trunk stimulation needs to be made crystal clear; in many places, "stimulus application" is used in places where this reviewer suspects they actually mean sympathetic nerve trunk stimulation.

We realize that this is confusing, and we also agree that at least in one place, we have not been sufficiently clear about the distinction. To clarify, we distinguish between stimulus application (physical application of stimulus to the nostril), and stimulation (which refers to SNT stimulation, which typically induces VNO suction). The general term stimulus presentation refers to the entire process. As explained in the text, in our analysis, we consider the entire window starting at application and ending 40s after stimulation. This is because we sometimes observe immediate responses following application. One such responses is seen in Figure 2D, and this is directly related to a detailed comment made below (on Figure 1D, part c). Indeed, for this figure time 0 indicates stimulus application. This was indicated previously, but we have now rearranged order of the panels to make the distinction between this response and other clearer. We have also revised the figure caption and the text to clarify this issue.

b) There appears to be a mismatch between the discussion of Figure 3 and its contents. Specifically, there is an example of an "adjusted" pattern in 3A, not 3B.

True. we have revised the text to correctly refer to the figure. Thanks.

c) The discussion of patterns neglects to mention whether it's possible for a neuron to belong to more than one pattern. For example, it would seem possible for a neuron to simultaneously fit the "ICR pattern" and the "dominant adjusted pattern" if, e.g., all ICR responses are stronger than all others, but if simultaneously within each strain the dominant male causes the largest response.

This is true. In the legend to Figure 3B, we actually wrote: “A neuron may fulfill more than one pattern and thus may appear in more than one row.”, but we now also write in the main text:

“We note that criteria for adjusted patterns are less stringent than for the standard patterns defined above. Furthermore, some patterns are not mutually exclusive, and thus, a neuron may fulfil more than a single pattern.”

(3) Discussion:a) The discussion of chemical specificity in urine focuses on volatiles and MUPs (citation #47), but many important molecules for the VNS are small, nonvolatile ligands. For such molecules, the corresponding study is Fu et al 2015.

Agreed. We now cite this work and several others that were not included before in the context of chemical and electrophysiological analyses.

b) "Following our line of reasoning, this scarcity may represent an optimal allocation of resources to separate dominant from naïve males": 1 unit out of 215 is roughly consistent with a single receptor. Surely little would be lost if there could be more computational capacity devoted to this important axis than that? It seems more likely that dominance is computed from multiple neuronal types with mixed encoding.

We fully agree, and we are not claiming that dominance, nor any other feature, is derived using dedicated feature selective neurons. Our discussion of resource allocation is inevitably speculative. Our main point in this context is that a lack of overrepresentation does not imply that a feature is not important. As a note, we do not think that there is good reason to suppose that AOB neurons reflect the activity of single receptors.

To present this potential confusion, we now added the following sentences in the Discussion subsection titled “Response patterns of AOB-MCs”:

“We stress that we do not suggest that features such as physiological state are encoded by the activity of single neurons. In fact, we believe that most ethologically relevant features are encoded by the activity of multiple neurons. Nevertheless, such population level representations ultimately depend on the response properties of individual neurons, and we thus ask: what can we learn from our analysis of response pattern frequency?”

(4) Methods:a) Male status, "were unambiguous in most cases": is it possible to put numerical estimates on this? 55% and 99% are both "most," yet they differ substantially in interpretive uncertainty.

Upon reexamination, we realized that this sentence is incorrect. Ambiguous cases were not considered as dominant for urine collection. We only classified mice as dominant if they “won” the tube test and exhibited dominant behavior in the subsequent observation period in the cage. The phrasing has now been corrected in the manuscript (Methods section).

b) Surgical procedures and electrode positioning: important details of probes are missing (electrode recording area, spacing, etc).

This information has been added to the Methods subsection “Surgical procedures and electrode positioning”

c) Stimulus presentation procedure: Are stimuli manually pipetted or delivered by apparatus with precise timing?

They are delivered manually. This has now been clarified in the text.

d) Data analysis, "we applied more permissive criteria involving response magnitude": it's not clear whether this is what's spelled out in the next paragraph, or whether that's left unspecified. In either case, the next paragraph appears to be about establishing a noise floor on pattern membership, not a "permissive criterion."

True, the next paragraph is not the explanation for the more permissive criteria. The more permissive criteria involving response magnitude are actually those described in Figure 3A and 3B. The sentence that was quoted above merely states that before applying those criteria, we had also searched for patterns defined by binary designation of neurons as responsive, or not responsive, to each of the stimuli (this is directly related to the next comment below). Using those binary definitions, we obtained a very small number of neurons for each pattern and thus decided to apply the approach actually used and described in the manuscript.

To clarify this confusion, we thoroughly derived the description of this paragraph, and the beginning of the next one in the Methods section.

e) Data analysis, method for assessing significance: there's a lot to like about the use of pooling to estimate the baseline and the use of an ANOVA-like test to assess unit responsiveness.But:i) for a specific stimulus, at 4 trials (the minimum specified in "Stimulus presentation procedure") kruskalwallis is questionable. They state that most trials use 5, however, and that should be okay.

The exact values are now given in the text. The mean number of repeated presentations per stimulus: 5.1± 0.9, mean ± sd. In 72% of the cases, stimuli were given 5 or more times. Otherwise, they were presented 4 times. In the context of the statistical test, we note that we are not comparing 5 (or 4) values with another set of 5 (or 4 values), but with a much larger sample (~44-55 baseline trials – given 11 trials and 4-5 repeats of each). Under this scenario, we think that the statistical approach is sound. However, the more important consideration, in our opinion, is given below.

ii) the methods statement suggests they are running kruskalwallis individually for each neuron/stimulus, rather than once per neuron across all stimuli. With 11 stimuli, there is a substantial chance of a false-positive if they used p < 0.05 to assess significance. (The actual threshold was unstated.) Were there any multiple comparison corrections performed? Or did they run kruskalwallis on the neuron, and then if significant assess individual stimuli? (Which is a form of multiple-comparisons correction.)

First, we indeed failed to mention that our criterion was 0.05. This has been corrected, by adding the information to the results and the Methods sections. No, we did not apply any multiple comparison measures. We consider each neuron-stimulus pair as an independent entity, and we are aware that this leads to a higher false positive rate. On the other hand, applying multiple comparisons would be problematic, as the same number of stimuli used in different studies varies. Application of multiple comparison corrections would thus lead to different response criteria across different studies, which would be very problematic. This raises the almost philosophical question regarding the use of multiple comparisons (as well as one and two tailed tests), but practically, most, if not all of our conclusions involve comparisons across conditions. For this purpose, we think that our procedure is valid. More generally, while selection of responses according to significance has some obvious advantages, the decision to use any particular criterion is entirely arbitrary. Therefore, we do not attach any special meaning to the significance threshold used here. Rather, we think of it as a simple criterion that allows us to exclude weakly responding or non-responsive neurons, and to compare frequencies of neurons that fulfill this criterion, under different conditions and contexts.

**Recommendations for the authors:**

**Reviewer #2 (Recommendations for the authors):**
Results:"are represented more than represented by chance" seems to have a misplaced word

True. Thanks. Corrected.

Figure 1D:a) Indicate the meaning of the number that appears in the top left for each unit (10, 5, 40, 5, 5) (I'm guessing it's the vertical scale for the PSTH, but best to spell it out explicitly.)

This information has been added.

b) "The red vertical line indicates stimulus application": is it the application of the chemical stimulus or SNT shock?

Please see our answer to c

c) "For unit 2, time 0 indicate stimulus application, as in this case, responses began after stimulus application, prior to stimulation." First, the meaning of time 0 for the other units is not clearly specified (we infer that unit 2 is an exception, but we don't know what most of them mean). Second, it seems as if the response (?) to ICR naive begins even before stimulus application.

This issue was also mentioned above as the 2nd weakness raised by this reviewer. To explain the meaning of the red lines, and resolve this confusion, we revised the figure caption text to indicate that for all units (except the former unit 2) time 0 indicates SNT stimulation. We also changed the order of the unit examples, placing the former unit 2 in the rightmost position. It is true that for this unit, there is a firing rate change prior to stimulus application, which actually appears as rate attenuation following stimulus application. In this specific case, we consider this activity as “noise”, and note that this neuron-stimulus combination would not be classified as a response (since there is no consistent change across stimulus presentation).

As a note, while reviewing this figure, we noted an error. We have previously written that the ITI was 10 s, whereas it was actually 18 s long. This has been corrected in the Figure and in the text.

Figure 2B:"The mean error due to the reduced 2-D representation is 0.29 (arbitrary units)." This is unclear. MDS is often described in terms of % of variance explained, is that what this means? If so, the units are not arbitrary; otherwise, it's unclear whether specifying a value with arbitrary units adds any value.

This is a very good point, and we thank the reviewer for identifying this mistake. The units are not arbitrary! They are units of correlation distance. We now added a scale bar (a square) to panel 2B to indicate what a distance of 0.1. Following this comment, we also calculated the mean error in the original data, and noted the ratio between the mean absolute error (due to considering only two dimensions) and the mean original distances. We also now report the value of the first two eigenvalues. Specifically, we now write:

“Note that like all dimensionally reduced representations, the representation in Fig. 2B is an approximation. Here, the first two eigenvalues of account for 44.6% of the variance of the original distances (30.4% and 14.2%, respectively for the first and second dimension). Another way to evaluate the representation is via the mean error due to the reduced 2-D representation. Here, it is 0.29, whereas the mean of the original distances is 0.73.”

Figure 3A:a) There is a truncated label (or something) above the panel letter.

Thanks. Corrected. This was part of the “Figure” label

b) The graphic for the "adjusted pattern" also fits the criterion of the "pattern": for example, in the top row the activity for ICR is still higher than for any other stimulus, thus fulfilling the criterion of a "pattern" and not just an "adjusted pattern."

That was not our intention. An adjusted pattern does not necessarily fulfill the (non-adjusted) “pattern” (while the opposite is true). We have now revised the rightmost panel in figure 3A, adding both “&s” to indicate that all three conditions must be fulfilled, and in attempt for a more intuitive representation, applied a different background denoting stimuli with irrelevant responses. We also changed the terms in the legend within the panel, making them more accurate: (Thus, “strong activity” was changed to “stronger responses”). In addition, we revised the text and figure legends in attempt to better clarify these definitions.

Figure 3B:I'm assuming that the columns of the heatmap correspond to different urine stimuli, and that the color is normalized firing rate. But readers should not have to guess.

True, and agreed. We added legends to clarify this.

Figure 4B:The caption should mention that the pairwise measures are between the stimulus columns of panel A.

We revised the caption to indicate this. Note that we also added two additional panels to this figure.

Figure 5A&B:Instead of a multiple-comparisons correction, it seems likely to be better to use a 2-way ANOVA. At a minimum, the nature of the multiple-comparisons correction needs to be specified (many are conservative, but they differ in the extent of how conservative they are).

We now write in the text that we used a Bonferroni correction (this information previously appeared only in the caption). We also found an error in the caption. We previously wrote that we used a binomial exact test for both panels A and B. However, only the data in panel A was calculated with a binomial exact test. The data in panel B was calculated with a one-way ANOVA.

We now also applied a 2-way ANOVA to response magnitudes (i.e., panel B). We find a main effect of stimulus, but not of state, and no effect of interaction between the two. This is consistent with our previous analyses. This analysis is now included in the text. We thank the reviewer for this suggestion.

**Editor's note:**
Should you choose to revise your manuscript, if you have not already done so, please include full statistical reporting including exact p-values wherever possible alongside the summary statistics (test statistic and df) and, where appropriate, 95% confidence intervals. These should be reported for all key questions and not only when the p-value is less than 0.05 in the main manuscript.